# Lanthanide-doped MoS$_2$ with enhanced oxygen reduction activity and biperiodic chemical trends

Yu Hao [1,2], Liping Wang[1,2] ✉ & Liang-Feng Huang [1,2,3] ✉

Molybdenum disulfide has broad applications in catalysis, optoelectronics, and solid lubrication, where lanthanide (Ln) doping can be used to tune its physicochemical properties. The reduction of oxygen is an electrochemical process important in determining fuel cell efficiency, or a possible environmental-degradation mechanism for nanodevices and coatings consisting of Ln-doped MoS$_2$. Here, by combining density-functional theory calculations and current-potential polarization curve simulations, we show that the dopant-induced high oxygen reduction activity at Ln-MoS$_2$/water interfaces scales as a biperiodic function of Ln type. A defect-state pairing mechanism, which selectively stabilizes the hydroxyl and hydroperoxyl adsorbates on Ln-MoS$_2$, is proposed for the activity enhancement, and the biperiodic chemical trend in activity is found originating from the similar trends in intraatomic $4f$–$5d6s$ orbital hybridization and interatomic Ln−S bonding. A generic orbital-chemistry mechanism is described for explaining the simultaneous biperiodic trends observed in many electronic, thermodynamic, and kinetic properties.

Oxygen reduction reaction (ORR) is an electrochemical process reducing O$_2$ into H$_2$O, which plays significant roles in the fields of clean energy and corrosion[1]. As the cathode reaction in fuel cells, active enough ORR is required for efficient energy conversion[2], and one contemporary urgent task is replacing the noble-metal catalysts (e.g., Pt) with less expensive, sufficiently active, and durable candidate alternatives[3]. On the other hand, as a cathodic reaction readily occurring in regular oxic humid/aqueous conditions[4], the activated ORR on a surface spot will cause the electron loss and potential rise of surrounding materials (e.g., metal substrates), which tends to induce the galvanic-corrosion phenomena[1,5]. Thus, accurately understanding the ORR behaviors and relevant mechanisms is not only desired by the design of advanced electrocatalysts but also by the appropriate protection and long-lasting working of many functional nanodevices and coatings in realistic environments.

Earth-abundant molybdenum disulfide (MoS$_2$) is a typical two-dimensional material with great application potential in catalysis, optoelectronic devices, and solid lubrication due to its preferred structural stability, suitable band gap, and easy shearing[6,7]. Lanthanide-doped MoS$_2$ (Ln-MoS$_2$) recently has emerged as an important group of materials with the electronic and optical properties profoundly tuned by the Ln dopants[8,9], and many kinds of Ln-MoS$_2$ systems (e.g., Ln = Sm, Eu, Dy, Ho, Er, and Yb) have been successfully synthesized in experiments[10–15]. Ln dopants can introduce many defect states in the band gap of MoS$_2$, and the degenerate $4f$ multiplet orbitals of Ln dopants are split by the highly anisotropic local crystal field in MoS$_2$ matrix. These two electronic-structure mechanisms will lead to the photoluminescence emission of MoS$_2$ from the visible range to the near-infrared spectrum, including the telecommunication range at $1.55\,\mu m$[10,12], as well as to the improved electrical property of

[1]Key Laboratory of Marine Materials and Related Technologies, Zhejiang Key Laboratory of Marine Materials and Protective Technologies, Ningbo Institute of Materials Technology and Engineering, Chinese Academy of Sciences, 315201 Ningbo, China. [2]Center of Materials Science and Optoelectronics Engineering, University of Chinese Academy of Sciences, 100049 Beijing, China. [3]Research Center for Advanced Interdisciplinary Sciences, Ningbo Institute of Materials Technology and Engineering, Chinese Academy of Sciences, 315201 Ningbo, China. ✉e-mail: wangliping@nimte.ac.cn; huangliangfeng@nimte.ac.cn

Ln-MoS$_2$[11,15], making Ln-MoS$_2$ promising for optoelectronic materials and nanodevices.

Pristine MoS$_2$ has a quite inert basal plane for ORR catalysis, and only small MoS$_2$ nanoflakes with a considerably increased ratio of active edge sites can exhibit observable ORR activity[16–18]. However, MoS$_2$ edges have low chemical stability and may incur degrading corrosion and oxidization of nanoflakes when exposed to realistic environments[19,20], thus large-scale MoS$_2$ flakes should still be preferred for long-lasting performance. Due to the significant tuning effect on the electronic structure of MoS$_2$, Ln doping may be a promising way to stimulate the ORR activity on its surface for catalysis purposes. However, for the nanodevices and lubricating films made of Ln-MoS$_2$, the activated ORR processes tend to bring unexpected galvanic-corrosion phenomena to many surrounding component materials (e.g., metal substrates and connecting wires)[1,19,21]. Therefore, it is meaningful and urgent to clearly understand and accurately predict the ORR behaviors on Ln-MoS$_2$ surfaces, which can not only motivate their future electrocatalytic applications but also guide the appropriate protection against galvanic corrosion for the long-lasting service of advanced nanodevices and coatings.

In this work, by considering all the 15 Ln dopants in MoS$_2$ and combining density-functional theory (DFT) calculations and current-potential polarization curve simulations, we discover the considerably enhanced ORR activity on Ln-MoS$_2$ surfaces with an intriguing modulating biperiodic chemical trend. We first use DFT to calculate the stability of Ln dopants in MoS$_2$, the adsorption stability of various ORR intermediates, and their reaction behaviors at the Ln-MoS$_2$/water interfaces, where the water effect is strictly modeled by statistically sampling the H$_2$O-film configurations. Many closely correlated biperiodic chemical trends are observed in various thermodynamic and kinetic properties, and the unifying electronic-structure mechanisms are revealed by analyzing the intraatomic orbital hybridizations and interatomic bondings of Ln dopants. A defect-state pairing mechanism is proposed for the selectively and largely (moderately) enhanced hydroxyl (hydroperoxyl) adsorption by Ln doping, which leads to the considerably enhanced ORR activity on Ln-MoS$_2$. We finally simulate the current-potential polarization curves for ORR processes on Ln-MoS$_2$ surfaces, where the individual roles of involved microkinetic steps are also clearly revealed.

## Results and discussion
### Biperiodic chemical trend in dopant stability

The screening of different Ln-dopant configurations in MoS$_2$ using DFT calculations (see section A in Supplementary Information, SI) reveals the most stable doping site located at the Mo site (Fig. 1a), which is consistent with the experimental observation using scanning transmission electron microscopy (STEM) (Fig. 1b)[10,15]. Furthermore, the calculated Raman spectra for such Ln-MoS$_2$ configuration also exhibit the same mode redshifts as the experimental measurements[10,22,23] (see section B in SI). These systematically close theory-experiment agreements strongly validate the atomic-structure model for Ln-MoS$_2$ constructed here, which will be used in the following calculations. The way of a material interacts with external environmental agents always largely depends on its intrinsic stability, and the stability of an Ln dopant in MoS$_2$ can be quantitatively described by its formation energy ($E_f$), which is defined as the energy change associated with the filling of a Mo vacancy in MoS$_2$ by a free Ln atom (see Eq. (1) in the "Methods" section) and then can directly reflect the atomic-bonding strength therein. The $E_f$s for all the 15 Ln dopants calculated using the standard Ln pseudopotentials (i.e., with valence $4f$ electrons) are shown in Fig. 1c, and the largely negative values ($-6$ – $-11$ eV) obviously indicate the high thermodynamic stability of Ln-MoS$_2$. In addition, the calculated phonon densities of states further prove the favorable dynamical stability of all the fifteen Ln-MoS$_2$ systems (see section A in SI).

In the variation of $E_f$ with respect to Ln type, we can observe a remarkable biperiodic chemical trend with a large modulating amplitude of 4.5 eV, which fully disappears if the Ln pseudopotentials with $4f$ electrons included in the ionic part are used in calculations (see the ionic-$4f$ $E_f$s in Fig. 1c), i.e., neglecting the participation of $4f$ orbitals in any interatomic bonding. From this dramatic difference between the valence-$4f$ and ionic-$4f$ $E_f$s, we can derive that although the $4f$ orbitals are highly localized (see section C in SI for atomic-orbital wavefunctions), the hybridization between $4f$ electrons with delocalized $5d$ and $6s$ electrons should play a significant role in many physicochemical properties of Ln-MoS$_2$. In addition to the above $E_f$s for Ln dopant in MoS$_2$, the Ln-dopant charge state ($q_{Ln}$, Fig. 1c) also exhibits a simultaneous biperiodic chemical trend. After a broader literature investigation, we further find more similar biperiodic trends in the Ln-metal sublimation heats, Ln-atom ionization potentials (IP), and homolytic

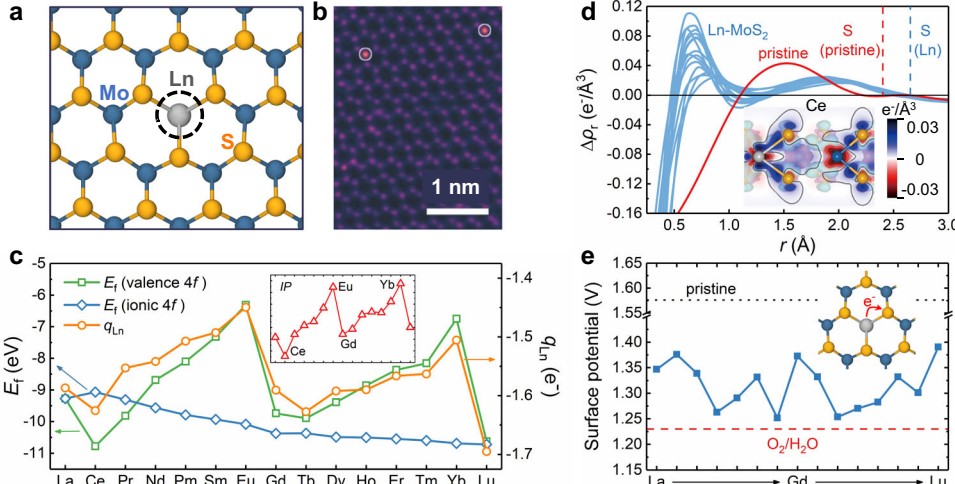

**Fig. 1 | The atomic structure, dopant stability, and related electronic-structure analyses for Ln-MoS$_2$. a** The atomic-structure model for Ln-MoS$_2$ used in this work and **b** the STEM image of Er-MoS$_2$ observed in the experiment, as reproduced with permission from ref. 10 (Copyright 2016, John Wiley and Sons). **c** The $E_f$s and $q_{Ln}$s for Ln dopants (inset: the weighted summation of $IP_3 + 0.2IP_4$). **d** The $\Delta\rho_r$ curves around the doping site ($r = 0$) for both MoS$_2$ and Ln-MoS$_2$ (inset: the distribution of $\Delta\rho(\mathbf{r})$ in Ce-MoS$_2$, side view), with the positions of S atoms labeled. **e** The surface potentials for both pristine MoS$_2$ and Ln-MoS$_2$ and the ORR equilibrium potential with respect to the standard hydrogen electrode (inset: the schematics for Ln–S electron transfer). Source data are provided as a Source Data file.

bond energies of LnF$_3$ molecules (see section C in SI)[24,25]. The weighted summation of the 4$f$-relevant third and fourth ionization potentials (IP$_3$ + 0.2IP$_4$) is also shown in Fig. 1c for comparison. Such generic biperiodic chemical trends in various properties of Ln elements in different states (e.g., atom, elemental metal, compound molecule, and solid dopant) can well validate our finding on the $E_f$s here and should be governed by a common intrinsic electronic-structure mechanism, which will be uncovered by analyzing the characters of valence 4$f$, 5$d$, and 6$s$ orbitals in the following.

The electronic wavefunctions and energy levels of Ln atoms in the $4f^{n_v-3}5d^16s^2$ configuration ($n_v$ is the valence-electron number) are calculated using the all-electron full-potential method[26], where highly localized (delocalized) character of the lower 4$f$ orbitals (upper 5$d$ and 6$s$ orbitals) clearly shows up (see Fig. S7a and S7b). The variation of magnetic moment in Ln-MoS$_2$ (see Fig. S7c) clearly proves such atomic configuration for Ln dopants, where the Hund's rule for magnetic 4$f$ electrons results in the monotonic increase (decrease) of the magnetic moment before (after) the half filling of 4$f$ orbitals, i.e., at Gd dopant with $4f^75d^16s^2$. The ground state of most free Ln atoms is in the $4f^{n_v-2}6s^2$ configuration, with the 4$f$ orbitals half filled at Eu atom ($4f^76s^2$), and the energy required for the $4f^{n_v-2}6s^2 \rightarrow 4f^{n_v-3}5d^16s^2$ transition on free Ln atoms also exhibits a biperiodic chemical trend as measured by experiments (see Fig. S7d)[27,28]: The transition energy increases in both the under-half-filled (from La to Eu) and over-half-filled branches (from Gd to Yb), which is bisected by an abrupt drop at Eu - Gd. Such biperiodic trend in electronic transition energy originates from a similar trend in the attractive exchange potential (origin for Hund's rule) felt by the 4$f$ electrons transiting up to 5$d$6$s$ orbitals. This also explains the biperiodic trends in ionization potentials of Ln atoms mentioned above. From the viewpoint of orbital chemistry[29], a lower 4$f$–5$d$6$s$ transition energy will lead to an easier 4$f$–5$d$6$s$ hybridization on an Ln atom, which can facilitate the stronger interatomic bonding of delocalized Ln-5$d$6$s$ orbitals with surrounding atoms, e.g., the Ln–Ln bonding in Ln metals, Ln–F bonding in LnF$_3$, and Ln–S bonding in Ln-MoS$_2$. Therefore, the biperiodic trends in intraatomic orbital hybridization and interatomic bonding can give out a unifying explanation for all the biperiodic trends in dopant stability of Ln-MoS$_2$ (Fig. 1c), sublimation heat of Ln metals (Fig. S5a), and homolytic Ln–F bond energy of LnF$_3$ (Fig. S6). For both La and Lu residing at the periodic-table-row ends, there exists a reverse 5$d$6$s$–4$f$ electron transfer in La-MoS$_2$, resulting in the decreased bonding 5$d$6$s$ electrons and then the upshifted $E_f$ (i.e., weakened dopant stability), while the increased number of 5$d$6$s$ electrons in Lu-MoS$_2$ leads to the lowered $E_f$ (i.e., strengthened dopant stability).

The Ln–S bonding strength as reflected by $E_f$ will closely correlate with many thermodynamic and kinetic quantities for the surface reactivity of Ln-MoS$_2$, because the bonding between an exterior adsorbate with an active S site is preceded by the endothermic partial breaking of the nearby Ln–S bonds. To clearly understand such interatomic bonding mechanism, the differential electron densities ($\Delta\rho$) induced by interatomic bonding are calculated for pristine MoS$_2$ and Ln-MoS$_2$, from which the radial distributions ($\Delta\rho_r$, by Eq. (2) in the "Methods" section) around the dopant site are further derived. The calculated $\Delta\rho_r$ curves for all the 15 Ln-MoS$_2$ systems are individually shown in Fig. S8 and summarized in Fig. 1d, where two common characters are prominent: (1) the accumulation of 4$f$ electrons at $r$ ~ 0.7 Å (i.e., $\Delta\rho_r > 0$) implying the 4$f$–5$d$6$s$ orbital hybridization and (2) the accumulated interatomic-bonding electrons at $r$ ~ 2.0 Å originating from the bonding between delocalized Ln-5$d$6$s$ orbitals and neighboring S-3$sp$ orbitals. The bonding electrons in the Ln–S bond of Ln-MoS$_2$ are closer to the S atom (by ~0.2 Å) than those in the Mo–S bond of pristine MoS$_2$, indicating the higher ionicity of Ln–S bond, and more electrons transferred out of the cation site after Ln doping. This can also be proved by the charge state of S atom (Fig. S9) and will be favored by the adsorption of ORR intermediates on S atom. It can be

derived that an easier 4$f$–5$d$6$s$ orbital hybridization on a Ln dopant will lead to more sufficient interatomic 5$d$6$s$–3$sp$ bonding and then more Ln–S electron transfer, which explains the simultaneous biperiodic chemical trends in both $E_f$ and $q_{Ln}$ (Fig. 1c). The above in-depth orbital analysis is consistent with the qualitative expectation from electronegativity values (Ln 1.00 ~ 1.25, Mo 2.15, and S 2.58)[24], and the dopant–matrix electron transfer decreases the surface potential from 1.58 V down to 1.25–1.39 V with respect to the standard hydrogen electrode (SHE, see more details in the "Methods" section) after Ln doping, closer to the ORR equilibrium potential of 1.23 V (Fig. 1e). It is also the similarity in Ln–S bonding character for all of the Ln-MoS$_2$ systems that allows us to select Ce- and Sm-MoS$_2$ as representatives in the following to analyze many calculated properties and mechanisms.

## Water effects for adsorbate stability

An ORR process mainly consists of the adsorptions and transitions of O$_2$, O, OH, and OOH intermediates (see section D in SI for more details), which can be understood by calculating their adsorption free energies ($\Delta G_{ads}$, see sections E and F in SI for detailed formula). Since ORR occurs at the solid/liquid interface, it is desired to accurately understand the effect of the water environment, and the dynamically accessible structures of H$_2$O molecules on MoS$_2$ should require sufficient statistical samplings (see the "Methods" section). This is especially necessary for the relatively weak (loose) MoS$_2$/water interface, as shown by a representative Ln-MoS$_2$/water interface in Figs. 2a and S10 (section G in SI), and the distance between water film and Ln-MoS$_2$ surface is around 2.1 Å (see Fig. S11 for detailed statistical analysis). Eighteen water structures are sampled from the molecular-dynamics simulations of 45,000 steps (0.5 fs/step), and many sampled atomic structures for Ln-MoS$_2$/water interfaces with and without adsorbates are shown in Figs. S12 and S13. Generally speaking, the water effect mainly includes three aspects: (1) setting up an electric field by forming the electrical double layer, (2) forming hydrogen bonds with the polar adsorbate and surface, and (3) bringing the endothermic reorientation process during a reaction. According to the classical double-layer theory[30], the electric field at a solid/water interface is about 10$^9$ V/m, which changes the $\Delta G_{ads}$s only by $\lesssim 0.02$ eV here (Fig. S14). On the representative Ce-MoS$_2$ surface, it can be seen that the statistical fluctuations in $\Delta G_{ads}$s (Figs. 2b and S15) have been well captured by the samplings here, which allows us to implement the Weibull-distribution analyses on the $\Delta G_{ads}$ data (Figs. 2c and S16). Then, the maximum-probability $\Delta G_{ads}$ for each kind of adsorbate is located and used as the statistically average $\Delta G_{ads}$, and the effect of interfacial hydrogen bonds can be revealed by comparing the average $\Delta G_{ads}$s with and without water film.

On Ce-MoS$_2$ surface, the $\Delta G_{ads}$s of OH and OOH (Figs. 2b and S15) are decreased by 0.29 and 0.14 eV, respectively, due to the existence of water film, where the stabilizing interfacial hydrogen bonding should have competed over the endothermic water reorientation. However, the $\Delta G_{ads}$s of less polar O and nonpolar O$_2$ are increased by 0.1 and 0.2 eV, respectively, due to the dominating effect of water reorientation. The changes in $\Delta G_{ads}$ ($\Delta\Delta G_{ads}$) caused by the water effect for O, OH, and OOH under different water structures are plotted against the corresponding hydrogen-bond length ($d_{H-bond}$) in Fig. 2d, where a near-logarithmic relationship shows up. The strong interfacial hydrogen bonds on adsorbed OH and OOH (labeled as OH* and OOH*) result in the short $d_{H-bond}$s and the sharp decrease of $\Delta\Delta G_{ads}$ with decreasing $d_{H-bond}$, while the weak hydrogen bonds on O* in the longer $d_{H-bond}$s and a flat variation of $\Delta\Delta G_{ads}$. The contribution of water reorientation can be uncovered by comparing the $\Delta G_{ads}$ values with a water film and a single H$_2$O molecule (Figs. 2b and S15), where the water-reorientation effect is absent in the later situation. The water-reorientation effect for O*, OH*, O$_2$*, and OOH* on Ce-MoS$_2$ are calculated to be 0.17, 0.16, 0.17, and 0.25 eV, respectively, which are

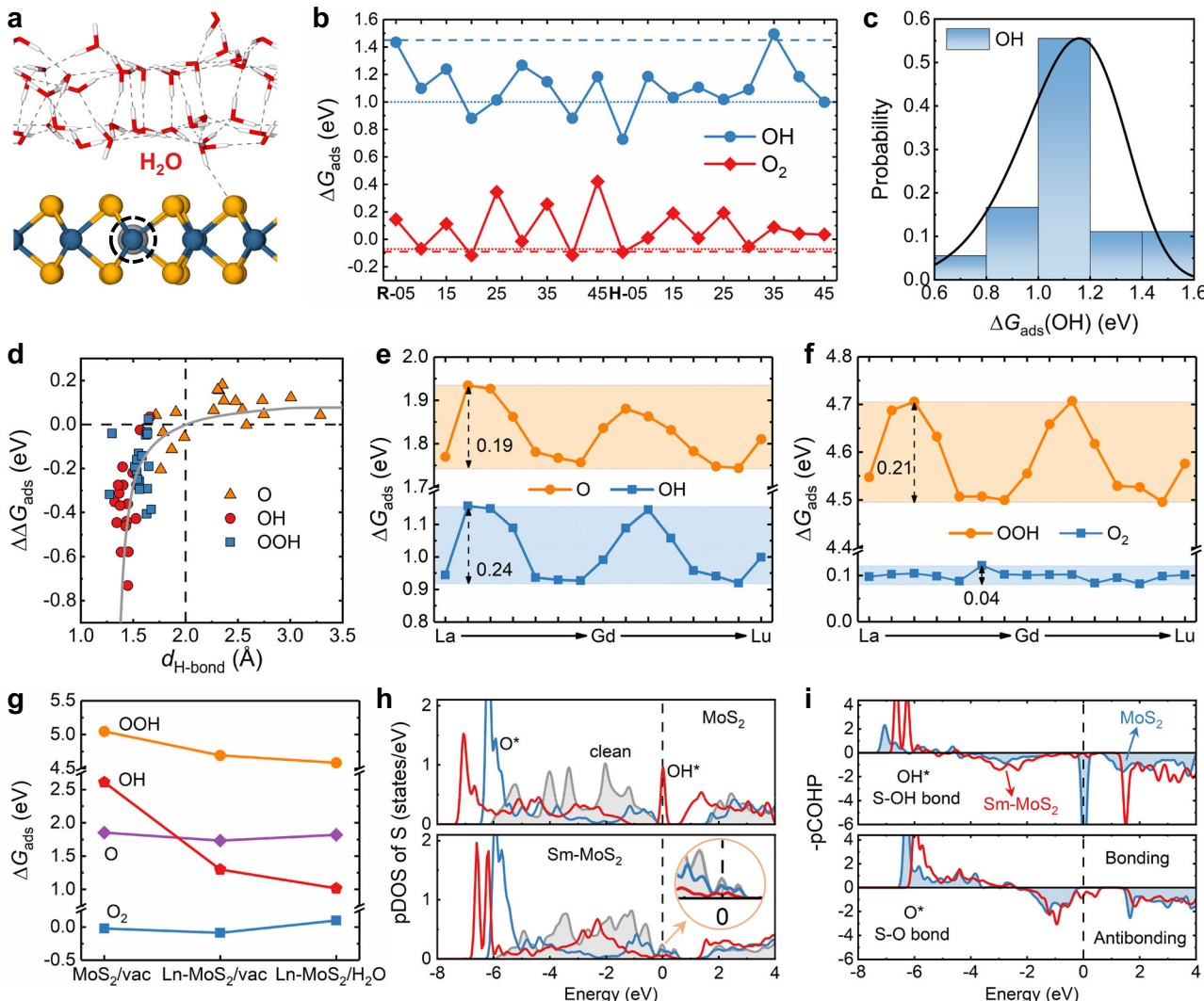

**Fig. 2 | The structural analysis of the Ln-MoS₂/water interface, adsorption-free energies of ORR intermediates, and related electronic-structure analyses. a** An example structure for the Ln-MoS₂/water interface. **b** The calculated $\Delta G_{ads}$s for OH and O₂ on Ce-MoS₂ with the statistically sampled water configurations, comparing with the results without water (dashed line) and with a single H₂O molecule nearby (dotted line). **c** The Weibull-distribution analysis for $\Delta G_{ads}$(OH) data. **d** The correlation between $\Delta\Delta G_{ads}$ and $d_{H-bond}$ data. **e, f** The $\Delta G_{ads}$s for O, OH, O₂, and OOH on Ln-MoS₂ surfaces with water. **g** The average $\Delta G_{ads}$s of ORR intermediates on different surfaces. **h** The pDOS spectra of the active S sites in pristine MoS₂ and Sm-MoS₂ (with/without O* and OH*) and **i** the –pCOHP spectra for the S–O bonds after O and OH adsorptions (0 eV: the highest occupied level). Source data are provided as a Source Data file.

the same for other similar Ln-MoS₂/water interfaces but are lower than those (0.22 – 0.28 eV) for Pt(111) surface with a stronger binding with water[31,32].

### Enhancement and biperiodic trends in surface adsorption

The $\Delta G_{ads}$s of O, OH, OOH, and O₂ on all the 15 kinds of Ln-MoS₂ surfaces underwater film are shown in Fig. 2e, f, where the simultaneous biperiodic chemical trends can be observed in the $\Delta G_{ads}$s for O, OH, and OOH that form covalent bonds with the substrate S atom. These biperiodic chemical trends in $\Delta G_{ads}$ are almost opposite to that in $E_f$ (Fig. 1c) because a weaker Ln−S bond (higher in $E_f$) is always easier to be perturbed by an exterior adsorbate (lower in $\Delta G_{ads}$). There are also well-defined linear relationships between the $\Delta G_{ads}$s of O, OH, and OOH (Fig. S17) because their adsorption stabilities rely on the S−O covalent bonding and then are tuned by the Ln dopant at the same pace. The biperiodic trend is absent in $\Delta G_{ads}$(O₂) that is determined by a weak electrostatic attraction between the adsorbate and surface, but such weak adsorption is still stronger than that on pristine MoS₂ by 0.03−0.13 eV (see section H in SI, Fig. S18).

The individual effects of Ln doping and water environment on the stability of any ORR intermediate can be sequentially disentangled by comparing the $\Delta G_{ads}$s at different surface states, i.e., pristine MoS₂ in a vacuum, Ln-MoS₂ in a vacuum, and Ln-MoS₂ with water (Fig. 2g). All the 15 Ln-MoS₂ surfaces are averaged for each data point in Fig. 2g to reveal the general effects of Ln doping and water environment, and these two effects for different adsorbates on all kinds of Ln-MoS₂ surfaces are shown in Fig. S19. The $\Delta G_{ads}$s of O₂ and O is only decreased by 0.07 and 0.12 eV after Ln doping, respectively, and OOH* by a moderate magnitude of 0.35 eV. However, an exceptionally large decrease of 1.31 eV is observed in $\Delta G_{ads}$(OH), associated with an obvious shortening in S−O bond by 0.16 Å (Fig. S20). It is regularly expected that different ORR intermediates may be stabilized by a similar energy magnitude, due to their common dependence on the surface reactivity[33]. Thus, it is somewhat counterintuitive to observe such large stabilizing effect of Ln doping selectively on OH*, for which the electronic-structure analysis below will reveal a special defect-state pairing mechanism. It is the weak OH* on pristine MoS₂ that usually acts as the ORR-rate

bottleneck[16], and Nørskov et al.[33,34] have also proposed that an ideal ORR catalyst can be realized when $\Delta G_{ads}(OH)$ is higher than that of Pt(111) by 0–0.2 eV. The $\Delta G_{ads}(OH)$s for many Ln-MoS$_2$ surfaces (Ln = La, Pm, Sm, Eu, Gd, Er, Tm, Yb, Lu) exactly reside within this favored energy region (see Fig. S21).

To understand the selective and large enhancement on $\Delta G_{ads}(OH)$ by Ln dopant, the projected density of states (pDOS) of S atom before and after adsorption, as well as the crystal orbital Hamilton population (pCOHP) spectra[35,36] for the S–O bonds on adsorbed surfaces, are calculated. The pDOS and pCOHP spectra for both pristine MoS$_2$ and Sm-MoS$_2$ surfaces (clean or adsorbed with O*/OH*) are compared in Fig. 2h and i to reveal the underlying electronic-structure mechanism, and the spectra of other Ln-MoS$_2$ surfaces have the same characters as those of Sm-MoS$_2$ surfaces (see Figs. S22 and S23). It can be seen that the adsorbate–S bonding increases the bonding states (–pCOHP > 0) at the valence-band edges (−7.5 ~ −5.0 eV), and the electronic states will progressively convert into the antibonding type (–pCOHP < 0) around −5.0 and −2.3 eV for OH* and O*, respectively. Comparing the pDOS and –pCOHP spectra for MoS$_2$, Ln-MoS$_2$, and OH@MoS$_2$, it can be seen that both Ln doping and OH adsorption will create localized antibonding defect states around the Fermi level. The selectively and largely enhanced adsorption of OH (with a single dangling bond) can be ascribed to the effective pairing between these two kinds of defect states. For the surfaces with chemically adsorbed OOH (with a single dangling bond), the pDOS spectra present the same characters as those of OH (Fig. S22) due to the same defect-state pairing mechanism. It should be noted that the decrease in $\Delta G_{ads}$ by Ln doping is less for OOH than OH (Fig. 2g) because the physical adsorption state of OOH (see Fig. S20) is more stable than its chemical state on pristine MoS$_2$, and then instead is used here to yield a dopant effect smaller than that of OH. In contrast, the adsorption of O (with double dangling bonds) does not create such an unpaired defect state on pristine MoS$_2$, thus the defect-state pairing mechanism is absent here. The adsorption strength can be also reflected by the integrated –pCOHP for the occupied valence states, and the obtained values for the S–O bonds in OH@MoS$_2$ and OH@Ln-MoS$_2$ are 5.0 and 7.1–7.3 eV, respectively, but both ~10.7 eV in O@MoS$_2$ and O@Ln-MoS$_2$. This quantitatively proves the defect-state pairing mechanism for the selective and large enhancement of OH* above, which can provide a precise chemical approach for the atomistic design of electrocatalysts in the future.

### Thermodynamic rationale for ORR activity

ORR mainly has two possible pathways, i.e., the O$_2$-dissociative and the OOH-associative ones (see section D in SI for a detailed description). On Ln-MoS$_2$, the dissociation of O$_2^*$ requires a quite high activation energy of ~1.4 eV and is difficult to overcome at room temperature. However, the associative transition of O$_2^*$ into OOH* only needs an activation energy of ~0.2 eV, because there exists the preferred attraction between a hydronium ion and the negatively charged O$_2^*$ (see Fig. S24). Similar mechanism has been also found on N-doped graphene, an excellent ORR catalyst realized in experiment[37]. It is indispensable to have a conductive surface to freely exchange electrons during an ORR process. As seen from the pDOSs of Ln-MoS$_2$ (Figs. 2h and S22), the defect states at the Fermi level brought by Ln doping indeed will result in the p-type conductivity of MoS$_2$, which is consistent with a recent experimental result from field-effect measurement on Sm-MoS$_2$[11]. In addition, the electron transfer from Ln dopant onto MoS$_2$ matrix and the lowered surface potential as revealed above (Fig. 1e) can promote more electrons transferred onto O$_2^*$ and then contribute to the enhanced ORR.

The ORR reactivity along the preferred association pathway can be well indicated by the corresponding free energy diagram (FED) at the ORR equilibrium potential of 1.23 V (see section E in SI for calculation formula)[38]. The reversible hydrogen electrode (RHE) is used as the default potential reference in this work unless otherwise specified.

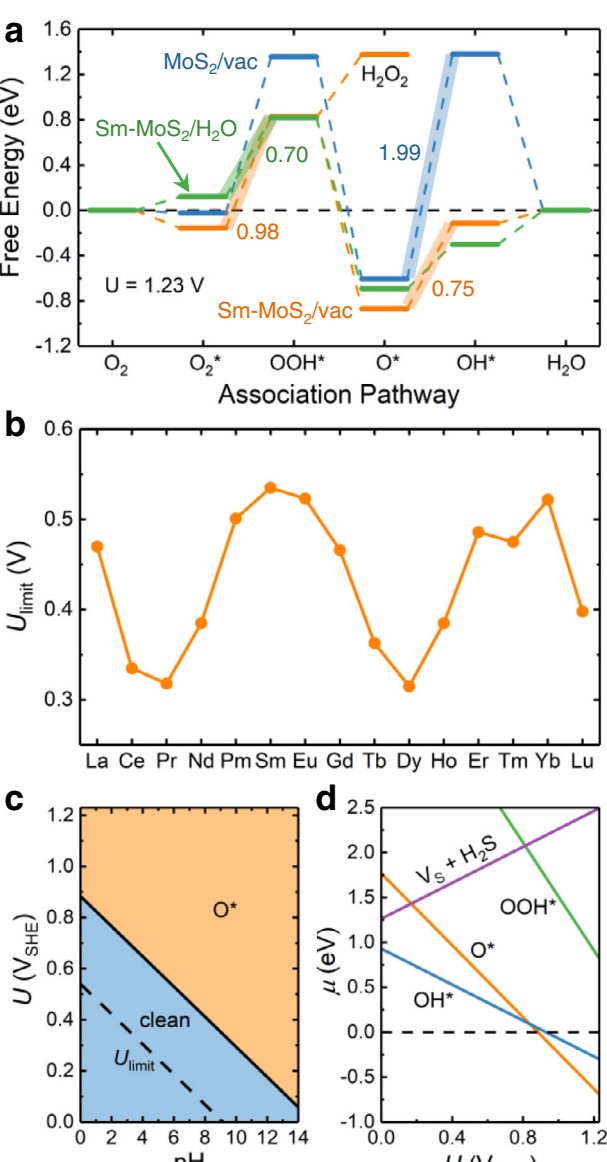

**Fig. 3 | Thermodynamic analyses for the ORR activity and adsorption state of Ln-MoS$_2$ surfaces. a** The FEDs (at $U = 1.23$ V) for the associative ORR pathway on pristine MoS$_2$ (without water) and Sm-MoS$_2$ (with and without water), **b** the variation of $U_{limit}$ with respect to Ln type, and **c**, **d** the surface Pourbaix diagram and involved potential-dependent chemical potentials ($\mu$, at pH = 0) on Sm-MoS$_2$. The FEDs, surface Pourbaix diagrams, and $\mu$s for other Ln-MoS$_2$ systems can be found in Figs. S25–S27, respectively. Source data are provided as a Source Data file.

In a FED at 1.23 V, the free-energy change associated with each step is defined as $\Delta G$, and the electron-involved step with the maximum-$\Delta G$ is the potential-limiting step for the whole ORR process. Below the corresponding limiting potential ($U_{limit} = 1.23 - \frac{\Delta G_{max}}{|e|}$), all ORR steps are exothermic. The FEDs for both pristine MoS$_2$ and Sm-MoS$_2$ are shown in Fig. 3a, and the FED profiles and potential-limiting steps for other Ln-MoS$_2$ surfaces are all the same as those of the representative Sm-MoS$_2$ surface (see section I in SI, Fig. S25). For the ORR on pristine MoS$_2$ in vacuum (i.e., neglecting water effect), the potential-limiting step is the protonation of O* into OH* with a very high $\Delta G$ of 1.99 eV. After Sm doping, the largely stabilized OH* leads to the considerably lowered $\Delta G$ down to 0.75 eV for this O* → OH* step. Then, the O$_2^*$ → OOH* step with a higher $\Delta G$ of 0.98 eV becomes the potential-limiting step. When the water effect is considered, the $\Delta G$s of these two steps on Sm-MoS$_2$ will further drop down to 0.39 and 0.70 eV, respectively, due to the

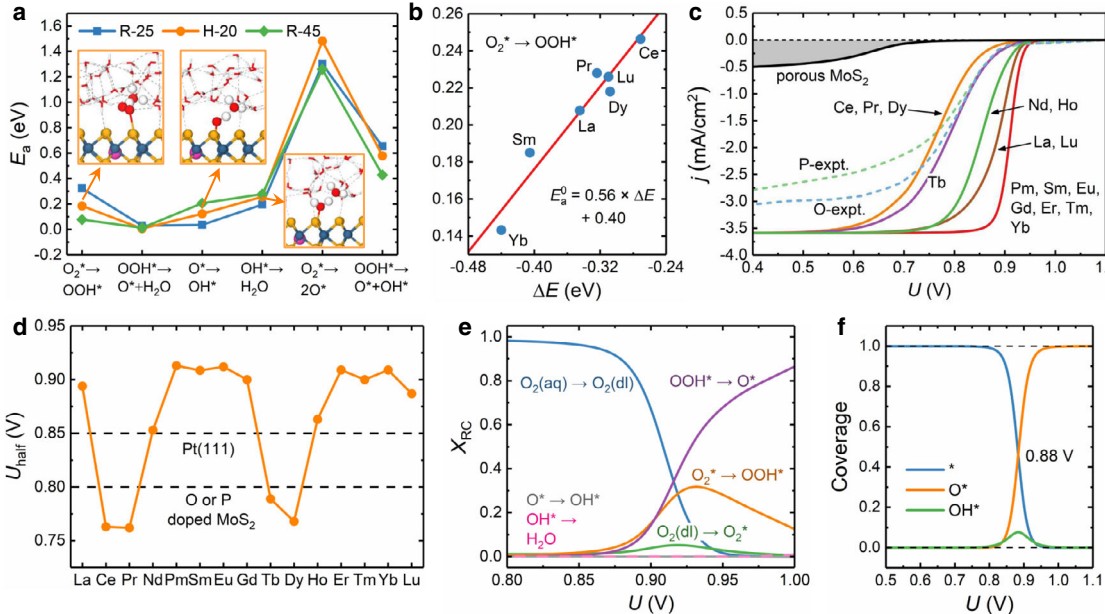

**Fig. 4 | The simulated and analyzed results for polarization curves. a** The $E_a$s for various reactions on Sm-MoS$_2$ under three different water configurations and **b** the linear $E_a$–$\Delta E$ relationship for the $O_2^* \rightarrow OOH^*$ step. **c, d** The simulated polarization curves for Ln-MoS$_2$ surfaces at 25 °C (disc rotation at 1600 rpm) and the derived $U_{half}$s, and the experimental polarization curves for porous, O-, and P-doped MoS$_2$ samples[43, 44] are compared in panel (**c**). **e, f** The simulated $X_{RC}$ and coverage curves for Sm-MoS$_2$. Source data are provided as a Source Data file.

stabilizing effect of interfacial hydrogen bondings on OH$^*$ and OOH$^*$. The $U_{limit}$s for all the 15 Ln-MoS$_2$ surfaces underwater are shown in Fig. 3b, where it can be seen that the magnitude is modulated between 0.31 and 0.54 V by a biperiodic chemical trend, and is almost inverse to the trend in $\Delta G_{ads}$(OOH) (Fig. 2f). These $U_{limit}$s are a little lower than that of Pt (0.78 V)[38] and close to those of MoS$_2$ edges (~0.57 V)[16,17]. In addition, although the possible byproduct H$_2$O$_2$ often has a negative impact on ORR performance, we find it difficult to be produced on Ln-MoS$_2$, because the endothermic OOH$^*$ → H$_2$O$_2$ step ($\Delta G \sim 0.56$ eV) cannot compete with the exothermic OOH$^*$ → O$^*$ step ($\Delta G \sim -1.69$ eV).

The surface Pourbaix diagram (see section J in SI for calculation formula)[4,39] can be used to reveal the electrochemical stability of Ln-MoS$_2$ surface state. The diagram and associated chemical potentials ($\mu$) for the representative Sm-MoS$_2$ surface are shown in Fig. 3c, d, and the similar electrochemical results for all the 15 Ln-MoS$_2$ surfaces are summarized in Figs. S26–S28. In addition, the possible release of H$_2$S from defective sites of MoS$_2$[40] is also considered in our electrochemical simulation here, where an active S atom (bonding with the Ln dopant) is extracted out to form an H$_2$S molecule, leaving an S vacancy (V$_S$) behind. From the persistent large positive $\mu$(H$_2$S + V$_S$)s for all Ln-MoS$_2$ surfaces at 0 ~ 1.23 V$_{RHE}$ (Figs. 3d and S27), it is clear that the active S atoms are very stable against the formation of H$_2$S. It is interesting to observe that $\mu$(H$_2$S + V$_S$) also has a biperiodic chemical trend (Fig. S29) reverse to that of $E_f$ (Fig. 1c) because the weaker an Ln−S bond is (higher in $E_f$), the less energy cost to form H$_2$S. According to the surface Pourbaix diagrams, any adsorbate is metastable at a potential around $U_{limit}$ and then will not remain for too long time on the surface during an ORR process, and O$^*$ will only become stable at potential >0.88 V (RHE). If two of the three active S sites are occupied by O$^*$, there will be an inter-adsorbate repulsion of 0.07 eV, making the double-O$^*$ configuration less stable than single O$^*$. Therefore, the Ln-MoS$_2$ surfaces will be stably preserved during the ORR process, and it is valid to consider the single adsorbate as the ORR reactant here.

### Polarization-curve simulations for ORR

A thermodynamic model may underestimate the ORR activity, because it only yields the active condition with all the involved steps being exothermic. However, some endothermic reaction steps can be kinetically overcome in realistic conditions (e.g., at room temperature), and it actually is the kinetic activity that is directly associated with many experimental measurements, e.g., current–potential polarization curves. For example, the single-Fe-atom catalyst on N-doped graphene may be predicted to be ORR inactive according to its low $U_{limit}$ (0.25−0.43 V), however, the measured/simulated polarization curves clearly reveal its high ORR activity comparable with the standard Pt(111) surface possessing a high $U_{limit}$ at 0.79 V[41,42]. Furthermore, in many previous theoretical simulations of different material surfaces, certain simplified water configurations are frequently used[41] or the kinetic barriers for ORR steps on Pt(111) are simply borrowed[42]. However, the above analysis of water effects and the following kinetic results can show that it is highly desired to use the appropriate water-molecular structure for the Ln-MoS$_2$/water interface (with a loose morphology and a specific chemical character) into the simulation of kinetic processes. To accurately understand the ORR activity and guide future related experiments, microkinetic simulations for the multiple-step ORR processes on Ln-MoS$_2$ surfaces are carried out here to obtain the potential-dependent current densities ($j$)[34,41]. The activation energy ($E_a$) of each ORR step is first calculated to derive its reaction rate constant, and then the obtained rate constants of all the steps are used to solve the simultaneous reaction equations for the ORR on a rotating disk electrode (RDE)[31]. More theoretical details are given in the METHODS section below and the section K in SI.

The $E_a$s for various ORR steps on representative Sm-MoS$_2$ surface with three different water configurations (labeled as R-25, H-20, and R-45) are shown in Fig. 4a, and the corresponding atomic-structure evolutions for these steps are shown in Fig. S30. The adsorption process of a O$_2$ molecule in the double layer, i.e., O$_2$(dl) → O$_2^*$, is proved to be not the rate-limiting step (see Fig. S31 and the description above it) and its atomic-structure evolution is not shown here. It can be seen from Fig. 4a that the two dissociative steps of O$_2^*$ → 2O$^*$ and OOH$^*$ → O$^*$ + OH$^*$ have $E_a$s (about 1.4 and 0.5 eV) much higher than those of the competing associative steps of O$_2^*$ → OOH$^*$ and OOH$^*$ → O$^*$ + H$_2$O (about 0.2 and 0.03 eV), respectively. Then, the reaction rates of the former two dissociative steps at room temperature will be lower than

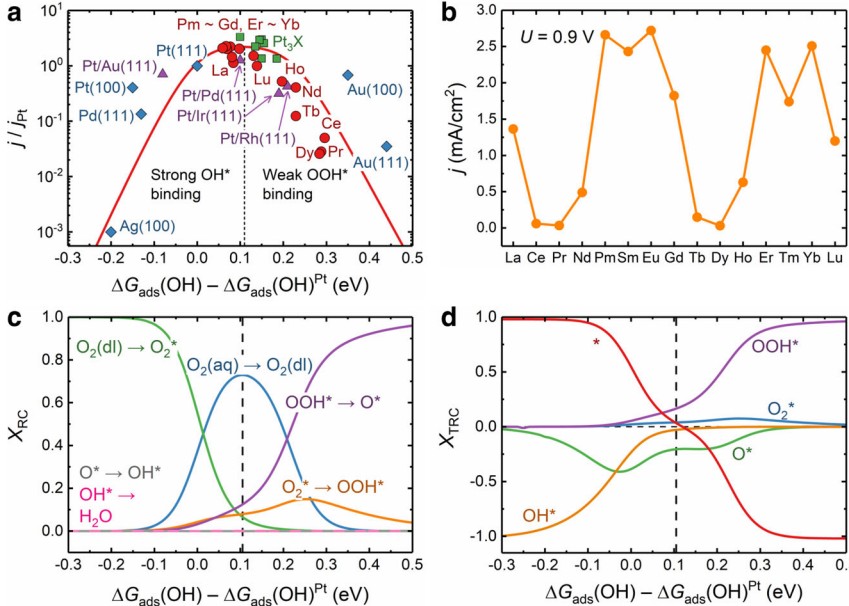

**Fig. 5 | The volcano plot and rate control analyses for the polarization current densities of Ln-MoS₂ surfaces. a** The volcano curve describing the $j$–$\Delta G_{ads}$(OH) relationship (at the standard 0.9 V) and the calculated results for all the fifteen Ln-MoS₂ surfaces, together with many reported results (experimental $j$ plus theoretical $\Delta G_{ads}$(OH)) for various noble-metal surfaces[34,38,60–69]. The $j$ and $\Delta G_{ads}$(OH) on Pt(111) surface is used as the references. **b** The variation of $j$ at 0.9 V with respect to Ln type. **c, d** The variations of $X_{RC}$ and $X_{TRC}$ curves with respect to $\Delta G_{ads}$(OH). Source data are provided as a Source Data file.

their counterpart associative steps by ≳ 10⁷ times and can be excluded from the possible ORR pathway. The H-2O water configuration is chosen to carry out the kinetic calculations for six more Ln-MoS₂ systems (Ln = La, Ce, Pr, Dy, Yb, Lu) due to the medium $E_a$ values yielded for Sm-MoS₂ (Fig. 4a). As shown in Figs. 4b and S32, S33, on these Ln-MoS₂ surfaces, the obtained $E_a$s for the O* → OH* step almost keep constant (-0.12 eV); the OOH* → O* + H₂O step has very low $E_a$s of ~ 0.03 eV; for the O₂* → OOH* and OH* → H₂O steps, their $E_a$s have the linear Brønsted–Evans–Polanyi relationships[31] with their reaction energies ($\Delta E$). These observed data tendencies are used to estimate the $E_a$s for the remaining eight Ln-MoS₂ systems, which can speed up the kinetic simulations here with the numerical accuracy safely guaranteed.

The simulated $j$–$U$ polarization curves for different Ln-MoS₂ surfaces at 25 °C and a typical RDE rotation speed (1600 rpm) are shown in Fig. 4c, where the curves indistinguishable from each other are merged together. More details for each $j$–$U$ curve, the derived onset potentials ($U_{onset}$), adsorbate coverages, the effect of RDE rotation speed (from 900 to 3200 rpm), and thermal effect (from 25 to 60 °C) can be found in Figs. S34–S38. The available experimental $j$–$U$ curves for porous MoS₂, O-MoS₂, and P-MoS₂[43,44] are also shown in Fig. 4c, and their similar potential dependence as those for Ln-MoS₂ surfaces can validate the microkinetic simulations here. As an effective kinetic indicator for ORR activity, the half-wave potential ($U_{half}$) tells the potential at which $j$ reaches half of its maximum value (i.e., the diffusion-limited value). The calculated $U_{half}$s for Ln-MoS₂ surfaces (Fig. 4d) exhibit a similar biperiodic chemical trend as that in $U_{limit}$ (Fig. 3b) and are close to or even higher than those of Pt(111), O-MoS₂, and P-MoS₂[31,43,44], indicating the superior ORR activity of Ln-MoS₂ surfaces. Furthermore, as another similar kinetic indicator, the derived $U_{onset}$s (Fig. S35) exhibit the same biperiodic trend and can also effectively reveal high ORR activity. The $U_{onset}$s are higher than the $U_{limit}$s by ~0.45 V, quantitatively indicating how far the kinetic activity is away from the thermodynamic threshold.

To fully understand the microkinetic mechanisms underlying the polarization curves, the degree of kinetic rate control ($X_{RC}$, see Eq. (6) in the "Methods" section)[45] is used to reveal the sensitivity of total ORR

rate ($r_{tot}$) to the rate-constant change of each step. The $X_{RC}$ curves for the representative Sm-MoS₂ surface are shown in Fig. 4e, where the dominating role of O₂ diffusion is replaced by the O₂* → OOH* and OOH* → O* steps at $U > 0.9$ V. Therefore, it is the fast forward transition through OOH* (a product in the potential-limiting step, Fig. 3a) that determines the ORR rate here. The OOH* → O* step ($E_a \lesssim 0.03$ eV) is almost spontaneous at room temperature, thus it actually is the O₂* → OOH* step with a secondary $X_{RC}$ value at $U = U_{half}$ that brings the biperiodic Ln-type dependence of ORR activity, and then it is the biperiodic chemical trend in $\Delta G_{ads}$(OOH) (Fig. 2f) that leads to the opposite trends in $U_{half}$ and $U_{onset}$. The simulated curves for surface-state coverages ($\theta$, in Figs. 4f and S36), are consistent with the aforementioned surface Pourbaix diagram (Fig. 3c), e.g., O* only becomes stable on the surface above 0.88 V and other metastable (kinetically important) intermediates have very low $\theta$s (<0.1).

## Volcano plot for ORR activity

Electrochemical reactivity is often understood and predicted by using the volcano plot[33], which well indicates that both too strong and too weak surface adsorptions will make ORR difficult to happen. Using the linear relationships between $\Delta G_{ads}$s of different ORR intermediates and the $E_a$–$\Delta E$ relationships discussed above, the analytical variation of $j$ within a given range of $\Delta G_{ads}$(OH) can be simulated, yielding the volcano curve as shown in Fig. 5a (red curve). The prototypical Pt(111) is considered as the reference surface in the volcano plot for setting the electrode potential (0.9 V, the $U_{onset}$ of Pt(111)), positioning the reference $\Delta G_{ads}$(OH) at 0.8 eV, and normalizing the $j$ values ($j_{Pt} = 1.2$ mA/cm²)[31,34]. The $j$–$\Delta G_{ads}$(OH) data for various noble-metal surfaces are also collected from literature for comparison in Fig. 5a, and their numerical data are listed in Table S3. According to the volcano-type curve, the highest $j$ can be obtained on a surface having a $\Delta G_{ads}$(OH) higher than that of Pt(111) by 0.0–0.2 eV, quantitatively consistent with the previous claim for metal surfaces[31,33,34]. Nine kinds of Ln-MoS₂ surfaces (Ln = La, Pm, Sm, Eu, Gd, Er, Tm, Yb, Lu) indeed reside in this optimal region with high $j$s (1.2–2.7 mA/cm²), and the other six kinds of surfaces (Ln = Ce, Pr, Nd, Tb, Dy, Ho) in the nearby region (relative $\Delta G_{ads}$(OH) at 0.2–0.3 eV) have moderate $j$s (0.03–0.6 mA/cm²). The

biperiodic chemical trend in $j$ can show up when plotted in terms of Ln type (Fig. 5b), which is in accordance with the biperiodic trends in other ORR-activity indicators, e.g., $U_{limit}$ (Fig. 3b), $U_{half}$ (Fig. 4d), and $U_{onset}$ (Fig. S35), but opposite to the biperiodic trends in $\Delta G_{ads}$s (Fig. 2e and f).

To reveal the microkinetic mechanisms underlying the volcano plot, we calculate the $X_{RC}$ curves in terms of $\Delta G_{ads}(OH)$ (Fig. 5c), as well as the curves for thermodynamic rate control ($X_{TRC}$, Eq. (7) in the "Methods" section)[46] to measure the sensitivity of $r_{tot}$ to the free-energy change of any surface state (Fig. 5d). A positive (negative) $X_{TRC}$ indicates that the increase in $r_{tot}$ needs to further stabilize (destabilize) the corresponding surface state. On the strong-adsorption side of the volcano plot (relative $\Delta G_{ads}(OH) < 0.0$ eV), the negative $X_{TRC}$s of OH* and O* indicate that their destabilization can lead to the increase in $r_{tot}$, while $X_{RC}$ is mainly dominated by the $O_2$ adsorption, because the more O atoms the surface captures, the faster a forward ORR reaction proceeds through the strong adsorbates (OH* and O*). On the weak-adsorption side (relative $\Delta G_{ads}(OH) > 0.2$ eV), the largely positive $X_{TRC}$ of OOH* indicates that it still needs to be stabilized for the increase in $r_{tot}$. This is the reason why the six Ln-MoS$_2$ surfaces (Ln = Ce, Pr, Nd, Tb, Dy, Ho) with the highest $\Delta G_{ads}$s have the lowest $j$s. In the optimal region (relative $\Delta G_{ads}(OH)$ at 0.0–0.2 eV), $r_{tot}$ is mainly determined by the $O_2$ diffusion in water and becomes almost surface-chemistry independent. This is the reason why various materials (e.g., doped MoS$_2$ and noble metals) with dramatically different chemical characters have very close $j$ values at the volcano top.

## Summary remarks for this study

In summary, we have carried out DFT calculations and polarization curve simulations for the ORR process on all the 15 Ln-MoS$_2$ surfaces. We not only have found the considerably enhanced ORR activity of MoS$_2$ surface induced by Ln doping, but also have identified a modulating biperiodic chemical trend in ORR activity with respect to Ln type. Many simultaneous biperiodic chemical trends have also been observed in various electronic structures, thermodynamic, and kinetic quantities, e.g., dopant stability, dopant charge state, ORR-intermediate adsorption strength, free energies of reaction for ORR intermediates (and $U_{limit}$), characteristic potentials for polarization curve ($U_{half}$ and $U_{onset}$), and current density. Based on the electronic-structure analysis, we find that the high ORR activity on Ln-MoS$_2$ originates from a defect-state pairing mechanism that selectively strengthens the hydroxyl and hydroperoxyl adsorptions, and the simultaneous biperiodic chemical trends originate from the similar biperiodic trends in intraatomic $4f$–$5d6s$ orbital hybridization on Ln dopant and interatomic Ln–S bonding. These analysis results also allow us to establish a generic orbital-chemistry mechanism that can closely correlate those simultaneous biperiodic trends in different properties. The ORR behaviors and key fundamental mechanisms revealed on Ln-MoS$_2$ can well guide more investigation and design of related material systems for many technologically important applications, e.g., electrocatalysts, optoelectronic nanodevices, and lubricating coatings.

## Methods
### DFT parameters and formula

DFT calculations are carried out using the VASP code package[47], where the ionic potential is described by the projector augmented-wave method[48]. The electronic exchange-correlation potential is expressed by the spin-polarized PBE functional in the generalized-gradient approximation (GGA)[49], and the dispersive van de Waals force is described using the zero-damping DFT-D3 functional[50]. The valence configurations in the used Mo, S, O, and H pseudopotentials are $4d^55s^15p^04f^0$, $3s^23p^43d^0$, $2s^22p^43d^0$, and $1s^12p^0$, respectively, and those in the Ln pseudopotentials include $5s$, $6s$, $5p$, $5d$, and $4f$ orbitals. The plane-wave cutoff energy is set to 450 eV, and the convergence

thresholds for atomic force and electronic energy are 0.01 eV/Å and $10^{-5}$ eV, respectively. A periodic $4 \times 2\sqrt{3}$ rectangular supercell of MoS$_2$ layer (12.61 × 10.92 Å$^2$) with an interlayer vacuum spacing of 20 Å is constructed as the structural model, and its Brillouin zone is spanned by a reciprocal-point grid of $2 \times 2 \times 1$.

Reaction paths and activation energies are calculated using the climbing-image nudged elastic band (CI-NEB) method[51] with a force convergence threshold of 0.05 eV/Å. The protonation rate of a surface adsorbate is limited by the reaction at water/MoS$_2$ interface because a proton can quickly reach the electrical double layer due to its very low diffusion barrier in water (0.07–0.11 eV[52]). To model this rate-limiting interfacial step, an H atom is placed on a water molecule nearby the adsorbate to form an H$_3$O unit, and the relaxed structural model is used as the initial state for the CI-NEB path (see Fig. S30). Crystal orbital Hamilton population analysis as implemented in the LOBSTER code package[35,36] is used to study the bonding and antibonding characters of atomic bonds, and atomic charges are calculated using Bader charge analysis[53]. The effect of electronic self-interaction problem intrinsic in the GGA functional for Ln atoms is tested by using the *GGA plus HubbardU*$_{eff}$ method[54], and found negligible for surface adsorption on S atom (see Table S4 in SI for details). More testing calculations on the spin–orbit coupling effect, cutoff energy, reciprocal-point mesh, supercell size, and magnetic configurations are also comprehensively carried out (see section L of SI), which further stringently validate the structural model and computational parameters considered in this work.

The formation energy ($E_f$) of an Ln dopant in MoS$_2$ is defined as the energy change associated with the filling of a Mo vacancy by a free Ln atom, which is expressed as

$$E_f = \varepsilon_d - \varepsilon_0 - \mu_{Ln}, \qquad (1)$$

where $\varepsilon_d$ and $\varepsilon_0$ are the total electronic energies of Ln-MoS$_2$ and MoS$_2$ with a Mo vacancy, respectively, and $\mu_{Ln}$ is the electronic energy of an isolated Ln atom. With such a definition, the obtained magnitude in $E_f$ will have a direct correlation with the interatomic bonding strength in Ln-MoS$_2$, which is highly useful for exploring the orbital-chemistry mechanism in both dopant stability and surface reactivity here.

The radial electron density distribution ($\Delta \rho_r$)[55] is calculated by

$$\Delta \rho_r(r) = \frac{1}{4\pi r^2} \iint_{|\tilde{\mathbf{r}}| = r} \Delta \rho(\tilde{\mathbf{r}}) d\sigma, \qquad (2)$$

where $\Delta \rho_r(r)$ is the average electron density on a spherical surface with radius of $r$; $\tilde{\mathbf{r}}$ is the position vector with length of $r$, and $\Delta \rho(\tilde{\mathbf{r}})$ is the bulk electron density at $\tilde{\mathbf{r}}$ point; $\sigma$ is the spherical-surface area.

The surface potential shown in Fig. 1e is calculated by referring the surface work function ($\Phi$) to that of the SHE ($\Phi_{SHE}$), which is expressed as

$$U = \frac{\Phi - \Phi_{SHE}}{|e|}, \qquad (3)$$

where $\Phi_{SHE}$ is measured to be 4.44 eV in experiment[56].

### Water-structure statistical sampling

MoS$_2$ will form a relatively weak interface with water, thus the interfacial H$_2$O structure should have a high degree of dynamical disorder, which requires sufficient statistical samplings to accurately obtain the average water effect. This is different from some metals (e.g., Pt) that can form quite strong metal-water interfaces, leading to some stable ordered interfacial water configurations[57]. We exploit the ab-initio molecular dynamics (AIMD) method to simulate such a dynamically disordered H$_2$O environment on Ln-MoS$_2$, where a thick enough water film with 32 H$_2$O molecules (thickness ∼ 7 Å) is considered. It is thicker

than that of the electrical double layer at the solid-water interface (-3 Å[38]). Two kinds of seed water structures are provided to initialize two threads of MD simulations: (1) the H atoms in $H_2O$ molecules at the interface pointing to the surface S atom (labeled as "H water"), and (2) the interfacial $H_2O$ molecules randomly oriented (labeled as "R water"). The weak interfacial interaction can be well proved by the relatively large interface distance (about 2.1 Å) in the simulated structures (Fig. S11). The Nosé–Hoover thermostat[58] is used in the AIMD simulations at 300 K for 45,000 steps (0.5 fs/step). There is no structural damage on the Ln-$MoS_2$ substrates during the AIMD simulations, indicating the preferred dynamical stability of doped structures. We sample the simulated Ln-$MoS_2$/water structures every 5000 steps and label them as H-05, H-10, …, and H-45 (R-05, R-10, …, and R-45) for the H-water (R-water) group, for which the atomic structures are shown in Fig. S12. The calculated electrostatic potentials along the normal direction of the Ln-$MoS_2$/water structures also clearly exhibit a two-layered morphology (Fig. S10) that is well-known as a typical solvent character on solid surfaces[59].

### Microkinetic modeling

The simultaneous rate equations for an ORR process can be briefly summarized as

$$\frac{\partial \theta_n}{\partial t} = \sum_i \nu_{ni} r_i = \sum_i \nu_{ni} \left( k_i \prod_R \theta_R - k_{-i} \prod_P \theta_P \right), \quad (4)$$

where $n$ indexes the species, and $r_i$ is the reaction rate of an elementary ORR step ($i$) involving the species $n$; $\nu_{ni}$ is the stoichiometric coefficient of species $n$ in step $i$, where $\nu_{ni}$ is positive (negative) if the species $n$ is a product (reactant); $\theta_R$ and $\theta_P$ represent the coverages of involved reactants and products in step $i$, respectively; and $k_i$ and $k_{-i}$ are the forward and reverse rate constants of step $i$, respectively. Together with some necessary constraints (e.g., the conservation of total state number), the set of simultaneous rate equations can be solved at the steady state, which is described in detail in SI (section K). The turnover frequency of $O_2$ ($TOF_{O_2}$) equals the total net reaction rate ($r_{tot}$), and is used to derive the current density ($j$) by

$$j = 4e \cdot \rho \cdot TOF_{O_2}, \quad (5)$$

where 4 is the number of transferred electrons, and $\rho$ is the surface density of active sites.

The sensitivities of $r_{tot}$ to the rate-constant change of each step (e.g., $k_i$) and the free-energy change of each species (e.g., $G_n^0$) can be revealed by the degree of kinetic rate control ($X_{RC}$) and thermo-dynamic rate control ($X_{TRC}$), respectively, defined as[45,46]

$$X_{RC,i} = \frac{k_i}{r_{tot}} \left( \frac{\partial r_{tot}}{\partial k_i} \right)_{k_{j \neq i}, K_i} \quad (6)$$

and

$$X_{TRC,n} = \frac{1}{r_{tot}} \left[ \frac{\partial r_{tot}}{\partial \left( \frac{-G_n^0}{k_B T} \right)} \right]_{G_{m \neq n}^0, E_a}, \quad (7)$$

where $K_i$ is the equilibrium constant of step $i$; $k_B$ is the Boltzmann constant; and a small variation of 1.0% in both $k_i$ and $G_n^0$ is used for the calculations of partial derivatives.

### Data availability

The data supporting all the conclusions of this study are available in the paper and Supplementary Information. Source data are provided with this paper. Additional data related to this study may be requested from the corresponding authors. Source data are provided with this paper.

### Code availability

The VASP code package used in this work to carry out the DFT calculations can be accessible after a user license is authorized by the VASP company (https://www.vasp.at).

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

## Acknowledgements

The authors are sponsored by the National Natural Science Foundation of China (Grant No. U21A20127 and 22272192, L.W. and L.-F.H.), the Fund of Science and Technology on Surface Physics and Chemistry Laboratory (Grant No. XKFZ202101, L.-F.H.), the National Key Research and Development Project (Grant No. 2022YFB3402803, L.-F.H.), and the Natural Science Foundation of Ningbo City (Grant No. 2021J229, L.-F.H.). The Supercomputing Center at Ningbo Institute of Materials Technology and Engineering is also acknowledged for providing the computing resources.

## Author contributions

Y.H. and L.-F.H. designed and performed the DFT calculations and wrote the draft. All authors analyzed the results and revised the manuscript. L.W. and L.-F.H. acquired the research funds and were responsible for supervision.

## Competing interests

The authors declare no competing interests.
