## [Peer Review File · Nature Communications]

Activated Oxygen Reduction Reaction with Biperiodic Chemical Trends on Lanthanide-Doped MoS₂REVIEWER COMMENTS

Reviewer #1 (Remarks to the Author):

In the work entitled "Oxygen Reduction Reaction on MoS₂: Activity Enhancement and Biperiodic Chemical Trends Induced by Lanthanide Dopant", Hao et al. performed density functional theory (DFT) calculations and polarization curve simulations of oxygen reduction reaction (ORR) processes on 15 Ln-MoS₂ surfaces. They found not only that the ORR activity on MoS₂ surfaces is greatly enhanced by Ln doping, but also that the ORR activity varies with Ln type in a biperiodic chemical trend. The biperiodic chemical trend was also observed in the electronic structure, thermodynamic and kinetic quantities, such as dopant stability, dopant charge state, ORR-intermediate adsorption strength, reaction free energy (and U_{limit}) of ORR intermediates, characteristic potentials of polarization curves (U_{half} and U_{onset}), and current density. This is a very comprehensive theoretical study, and an interesting topic. Hopefully the author would publish it after solving the problems.

1. Typically, DFT is not capable to describe Ln elements accurately, it is recommended to provide the reliability of their predictions.
2. It was reported that spin coupling and magnetic coupling associates with the catalytic activity [J. Am. Chem. Soc. 2018, 140(45), 15149-15152; Adv. Powder Mater. 2022, 1(3), 100031], authors might also consider such coupling.
3. The authors' definition of the limiting potential as $U_{\text{limit}} = 1.23 - \Delta G_{\text{max}}/e$ is different from the definition in many literature, one example is [ACS Appl. Mater. Interfaces 2020, 12, 52549-52559]. Authors are suggested to carefully check it.
4. When evaluating the stability, the text says that the phonon spectrum was calculated "In addition, the calculated phonon spectra for Ln-MoS₂ (Figure S2) further prove the favorable dynamical stability of Ln-MoS₂.", but Figure S2 presents the phonon density of states instead of the phonon spectrum. Authors only provide three systems in Figure S2, instead, all the considered systems are suggested to be given.
5. Figure S20 shows that the potential limiting step for all the 15 catalysts is the first electron step, and from the equation of limiting potential versus free energy, it is clear that the limiting potential is inversely related to ΔG_{OOH} . Most of the catalysts in FIG. 3b correspond well with FIG. 2f, except for Pm, Sm, and Eu, please check the data.
6. The authors compared only the Sm-MoS₂ structure with the pristine MoS₂ in Figure S16, and derived the difference between the whole Ln-doped systems and MoS₂, which is not rational, please complete the calculations for other doped systems.
7. Similarly, FIG. 2h and i are calculated only for Sm-MoS₂ for comparison with MoS₂, which could not indicate universalism, please calculate the other 14 structures or remove the discussion of the other 14 structures in the ORR section.
8. The authors say that FIG. 2 is the average of all Ln-doped structures, authors might give the reason for such comparison.
9. The energy cutoff of 450 eV and kpoint of 2×2×1 might be tested for accuracy.
10. It is known that good conductivity benefits the electrochemical process, and MoS₂ monolayer is semiconductor, authors might exploit the band structures of the Ln-doped MoS₂.
11. As for the water effect, what is the distance between MoS₂ and water layer?

12. “valent electrons” should be “valence electrons”?

Reviewer #2 (Remarks to the Author):

This manuscript reported a Ln-doped MoS₂ with high ORR activity and discovered a biperiodic chemical trend with respect to Ln type. The relationship between the defect-state pairing mechanism and the corresponding ORR activity was studied in detail. However, the biperiodic chemical trend of lanthanide is known to us and has been thoroughly studied before, and the creativity of this work is yet to be further discovered. I can't recommend this manuscript to be published.

1. The biperiodic chemical trends of lanthanide has long been known, which have been demonstrate in previous experiments and calculations (Brewer, L., Systematics and the Properties of the Lanthanides. First ed.; D. Reidel Publishing Company: 1982; Phys. Chem. Chem. Phys., 2013,15, 7839-7847; Thermochemica Acta,1992, 209, 175-188).
2. Layered MoS₂ is not the mainstream candidate for ORR with a fairly inert basal plane. The significance of choosing MoS₂ as the model needs further clarification.
3. The particularity of Sm needs to be elucidated, why was it chosen as the ORR model?
4. The stability of Ln-doped MoS₂ is estimated by the formation energy. Have the phonon spectra been calculated to determine their dynamical stability? What is the doping concentration of the lanthanide? Whether this is consistent with the experiment. What is the active site of Ln-doped MoS₂, lanthanide, S or Mo? Whether the biperiodic chemical trend is affected by the lanthanide content?
5. Now that the authors have obtained the material structure (Fig. 1b), it is encouraged that the catalytic activity to be tested experimentally.
6. In Fig. 1d, the curves of different Ln doped MoS₂ need to be differentiated.
7. There are some clerical errors in the article, for example, in page 4, this is not figure 1f, but figure 1e.

Reviewer #3 (Remarks to the Author):

The authors present a computational investigation of lanthanide-doped MoS₂ (Ln-doped, where lanthanide atoms substitute for Mo atoms) as active sites for oxygen reduction reaction (ORR). The

authors show a biperiodic trend in the energetics of this system (ranging from the energy of doping to the binding energy of adsorbates) and kinetics of ORR across the lanthanide series. Interestingly, OH and OOH bind rather strongly on the Lanthanide doped sites (i.e. on the S atoms bound to the dopant) and this is the primary reason for the enhanced activity of ORR on these sites relative to MoS₂. The paper is technically strong and this reviewer particularly likes that the electrochemical analysis was based on a kinetic model (rather than based on the computational hydrogen electrode assessments alone). However, several aspects of this contribution remain unclear and are listed out below.

1. The first question relates to the importance and impact of this work which cannot be readily assessed. The predicted activity of the Ln-doped catalysts seems high compared to Pt(111), so these materials do sound promising. Are there any experimental reports supporting such high activity? The authors do not seem to allude to any literature that corroborates this. The authors also mention that studying ORR is relevant to understand the mechanism of corrosion in such materials but it is unclear how big a challenge this is (i.e. is this problem the main impediment in deploying these materials). Further, if this issue is an important motivator, then is the chosen model (water-2D material interface) the correct description of the reactive environment?

2. The biperiodic trend is the particularly interesting and intriguing result of this work. Is there any experimental evidence for such behavior across doped 2D materials, even if not Ln-MoS₂?

3. Are these sites stable during electrocatalysis? For instance, what are the energetics of the loss of S atoms bound to the dopant? In general, substitutional dopants can also weaken the metal-sulfur bonds which can then make it easy to lose the adjacent S atoms electrochemically or thermochemically. What are the energetics of the competing reaction $2H^+ + 2e^- + S \rightarrow H_2S + vac$?? Understanding this might be key to knowing if these catalysts will be stable.

4. The current density of the catalyst is related to the active site density of Ln-doped MoS₂. In this context, what is the motivation for considering the chosen supercell dimension? Is it based on any synthesis information? What is the maximum expected extent of Ln doping on MoS₂? Relatedly, if indeed, more Ln doping is possible than even considered in the calculations, how close can two dopant atoms get? A more substantive discussion of the doping density is needed to justify the high predicted activity on these catalysts.

5. As pointed out earlier, a strength of this work is the development of j-V plot computationally using a microkinetic model. The authors, however, have not discussed how exactly the barriers were calculated. From some images in the SI, it seems like the barriers are calculated using CI-NEB calculations of a proton transfer (or proton-coupled electron transfer) from a hydronium ion to the surface adsorbate (O, OH, OO, OOH). More details about these calculations are needed (e.g. how do they choose their initial states for the NEB, etc.)

6. The microkinetic model is pretty informative about the rate-determining steps and abundant surface intermediates and their variation across lanthanides. This reviewer recommends comparing the surface coverage from the model with the surface Pourbaix diagrams. Ideally, the authors could consider building a kinetic phase diagram and comparing it with the thermodynamic phase diagram to understand how far the system may be away from equilibrium.

7. It seems like, right now, the authors only consider reactions 1-6 in their microkinetic model. Why not consider all ten reactions given in the SI in the model and let it determine the flux-carrying pathways under various scenarios? Since some reactions are thermochemical (and all reactions are ultimately dependent on temperature), the model will also then enable an understanding of the effect of temperature on the ORR activity. This can be valuable to predict the impact of higher operating temperatures in an electrochemistry context or in the context of galvanic corrosion.

Reviewer #4 (Remarks to the Author):

This is an interesting study which can be improved, see (minor and major) comments below.

- The hyperlink to ref 19 (Büttgen et al.) is dead.
- Why is the weighted sum of ionization potentials shown in Figure 1c, and where do the coefficients 1 and 0.2 for IP3 and IP4 respectively, come from?
- The surface potential part in Methods B is wrong. It is not RHE, but SHE that can be related to the absolute electrode potential of 4.44 V. Then RHE is obtained by including the pH dependence $0.059 \cdot \text{pH}$.
- Even though it is stated in the Methods that RHE is used throughout the paper unless otherwise specified, this is at the very end. I suggest that it is mentioned the first time it is used in the text and referring to the Methods for the details.
- In Section B.1., O₂, O, OH, and OOH are mentioned as examples of ORR intermediates. Isn't that all of them?
- I would have preferred to have the energy diagrams in SI section I drawn at 0 V vs. RHE. I struggled to see the relation between U_{limit} and the largest step (though admittedly I am more used to electrooxidation reactions)
- In the SI, section J, there is only one degree of freedom for the Pourbaix diagrams on the RHE scale, so the data could be presented in a table with U_{limit} and UO*/clean equilibrium.
- In the SI, section K, it says "It can be seen that the dissociation energy barrier of O₂* is as high as 1.26 ~ 1.50 eV and then is difficult to occur at room temperature, ...". I suggest a change to "...and then is difficult to overcome at room temperature, ...". A bit further down, on page S-20, it says "The current density (j) can be deduced...". I would have written "...can be calculated...".
- At the end of page 9 and in the figure text for Figure 4b, it says "liner" but should say "linear". On page S-22, there are two occurrences of "liner", one in the main text and one in the text of Figure S24.
- In Figure S27, the label for the y axis is "Coverage (θ)" but the other y axis labels have the form

Quantity (Unit), so I would suggest to either remove the paranthesis or write something like “fraction of ML”.

- In Table S3, there is a capitalized “Catalysts” which should be “catalysts”.
- In SI, section L, it says “...S atom, for which the PBE-GGA functional is accurate...”. Could I have a source for that claim?

Response to the Reviewers

We would like to express our deep gratitude to all the reviewers for their great effort, precious time, and specialized comments, which have greatly helped us to modify the science and story present in our paper. Each review comment has been carefully addressed, together with many additional calculation results newly obtained during this round of revision. Please check the new versions of Manuscript and Supplementary Information, the following *Change Lists* for both files, the following point-by-point author replies, and the files of *marked_Manuscript_for_review.pdf* and *marked_SI_for_review.pdf* with changes marked in red color.

Change List for Manuscript (colored in red in the marked Manuscript):

Page 3, Figure 1c; Page 3, 1st paragraph; Page 3, 2nd paragraph; Page 4, 2nd paragraph. According to the **Comment (13)** of **Reviewer #1**, the word “valent” has been changed to be “valence”.

Page 3, Figure 1e. According to the **Comment (4)** of **Reviewer #4**, the caption has been modified.

Page 3, 1st paragraph. According to the **Comment (5)** of **Reviewer #1**, the expression “phonon density of states” has been used instead of “phonon spectra”.

Page 3, 2nd paragraph. According to the **Comment (1)** of **Reviewer #2**, the sentences of “the Ln-metal sublimation heats ... LnF₃ molecules (see section C in SI)” and “Such generic biperiodic chemical trend ... in the following.” have been modified.

Page 4, 2nd paragraph. According to the **Comment (1)** of **Reviewer #2**, the sentences of “e.g., the Ln–Ln bonding in Ln metals ... in Ln-MoS₂.”, and “Therefore, the biperiodic trends in intraatomic orbital hybridization and ... and homolytic Ln–F bond energy of LnF₃ (Figure S6).” have been modified.

Page 4, 3rd paragraph. According to the **Comment (7)** of **Reviewer #2**, the sentence of “The calculated $\Delta\rho_r(r)$ curves ... originating from the bonding between delocalized Ln-5d6s orbitals and neighboring S-3sp orbitals.” has been modified.

Page 5, 1st paragraph. According to the **Comment (4)** of **Reviewer #4**, the words of “with respect to the standard hydrogen electrode (SHE, see more details in METHODS section)” has been added.

Page 5, 1st paragraph. According to the **Comment (8)** of **Reviewer #2**, the correct “Figure 1e” has been used.

Page 5, 2nd paragraph. According to the **Comment (6)** of **Reviewer #4**, the sentence of “An ORR process mainly consists of the adsorptions and transitions of O₂, O, OH, and OOH intermediates” has been modified.

Page 5, 2nd paragraph. According to the **Comment (12)** of **Reviewer #1**, the sentence of “and the distance between water film and Ln-MoS₂ surface is around 2.1 Å (Figure S11)” has been modified.

Page 7, 2nd paragraph. According to the **Comment (9)** of **Reviewer #1**, the sentence of “All the fifteen Ln-MoS₂ surfaces ... are shown in Figure S19.” has been modified.

Page 7, 3rd paragraph. According to the **Comment (8)** of **Reviewer #1**, the sentence of “The pDOS and pCOHP spectra for both pristine MoS₂ and Sm-MoS₂ surfaces ... as those of Sm-MoS₂ (see Figure S22 and S23).” has been modified.

Page 8, Figure 3d. According to the **Comment (5)** of **Reviewer #3**, the chemical potential of the state V_S + H₂S has been added.

Page 8, 2nd paragraph. According to the **Comment (9)** of **Reviewer #4**, the word “overcome” has been used.

Page 8, 2nd paragraph. According to the **Comment (11)** of **Reviewer #1**, the sentences about the enhanced conductivity of Ln-MoS₂ have been added.

Page 9, 2nd paragraph. According to the **Comment (5)** of **Reviewer #4**, the sentence of “The reversible hydrogen electrode (RHE) is used as the default potential reference in this work unless otherwise specified.” has been modified.

Page 9, 2nd paragraph. According to the **Comment (4)** of **Reviewer #1**, the sentence of “In a FED at 1.23 V, the free-energy change ... all ORR steps are exothermic.” has been modified.

Page 9, 2nd paragraph. According to the **Comment (7)** of **Reviewer #4**, the sentence of “The FEDs for both pristine MoS₂ ... as those of the representative Sm-MoS₂ surface.” has been modified.

Page 9, 2nd paragraph. According to the **Comment (6)** of **Reviewer #1**, the sentence of “The U_{limitS} for all the fifteen Ln-MoS₂ surfaces ... the trend in $\Delta G_{\text{ads}}(\text{OOH})$ (Figure 2f).” has been modified.

Page 9, 3rd paragraph. According to the **Comment (5)** of **Reviewer #3**, the discussion about the chemical potential of the state $V_S + \text{H}_2\text{S}$ has been added.

Page 10, Figure 4f. According to the **Comment (7)** of **Reviewer #3**, the transition potential between O^* and clean surface has been labeled.

Page 10, Figure 4f. According to the **Comment (11)** of **Reviewer #4**, the label of the y axis has been modified.

Page 10, 2nd paragraph. According to the **Comment (6)** of **Reviewer #3**, the sentence of “The E_{aS} for various ORR steps ... are shown in Figure S30.” has been modified.

Page 10, 2nd paragraph. According to the **Comment (8)** of **Reviewer #3**, the sentences of “It can be seen that the two dissociative steps ... can be excluded from the possible ORR pathway.” have been modified.

Page 11, 2nd paragraph. According to the **Comment (8)** of **Reviewer #3**, the sentence of “More detailed information ... can be found in Figure S34-S38.” has been modified.

Page 11, 2nd paragraph. According to the **Comment (7)** of **Reviewer #3**, the sentence of “The U_{onsetS} are higher than the U_{limitS} by ... the thermodynamic threshold.” has been added.

Page 14, 1st paragraph. According to the **Comment (6)** of **Reviewer #3**, the details for the reaction-path calculations have been added.

Page 14, 1st paragraph. According to the **Comments (3)** and **(10)** of **Reviewer #1**, the sentence of “More testing calculations ... considered in this work.” has been added.

Page 14, 4th paragraph. According to the **Comment (4)** of **Reviewer #4**, the description about the surface potential has been modified.

Page 15, 1st paragraph. According to the **Comment (12)** of **Reviewer #1**, the sentence of “The weak interfacial interaction ... in the simulated structures (Figure S11).” has been modified.

In addition to the above modifications, the writing English has also been polished all through the Manuscript and Supplementary Information.

Change List for Supplementary Information (colored in red in the marked SI):

Page S-2, 2nd paragraph. According to the **Comment (5)** of **Reviewer #1**, the sentence of “The dynamical stability ... where no imaginary phonon mode appears.” has been modified.

Page S-3, Figure S2. According to the **Comment (5)** of **Reviewer #1**, the phonon densities of states for all the fifteen Ln-MoS₂ structures have been added.

Page S-6, 1st paragraph. According to the **Comment (3)** of **Reviewer #4**, the sentences of “Similar biperiodic trend is also observed ... compared with the variation trend in the formation energy of Ln-MoS₂ (see main text, Figure 1c).” have been modified.

Page S-6, 1st paragraph. According to the **Comment (1)** of **Reviewer #2**, the sentence of “Furthermore, the biperiodic trend also appears in the homolytic Ln-F bond energies ... of Ln in different states (e.g., atom, metal, and compound molecule).” has been modified.

Page S-7, Figure S6. According to the **Comment (1)** of **Reviewer #2**, the biperiodic trend in homolytic Ln-F bond energies of LnF₃ molecules have been added.

Page S-8, 1st paragraph. According to the **Comment (1)** of **Reviewer #2**, the sentence of “It is such intrinsic orbital mechanism ... and formation energies of Ln dopants in MoS₂ (see main text, Figure 1c).” has been modified.

Page S-9, Figure S8. According to the **Comment (7)** of **Reviewer #2**, the radial distributions of differential electronic densities for all the fifteen Ln-MoS₂ systems have been added.

Page S-13, Figure S11. According to the **Comment (12)** of **Reviewer #1**, the statistical analysis for the distance between Ln-MoS₂ surface and water layer has been added.

Page S-17, 1st paragraph. According to the **Comment (9)** of **Reviewer #1**, the sentences of “The ΔG_{ads} for different adsorbates on Ln-MoS₂ surfaces ... over all the fifteen Ln-MoS₂ systems, as shown in Figure 2g.” have been modified.

Page S-17, Figure S19. According to the **Comment (9)** of **Reviewer #1**, the effects of Ln doping and water environment on the ΔG_{ads} for different adsorbates on all the fifteen Ln-MoS₂ systems have been added.

Page S-18, 1st paragraph. According to the **Comment (7)** of **Reviewer #1**, the sentences of “The adsorption structures and the differential electronic densities ... indicating the stronger covalent S-O bonding on Ln-MoS₂.” have been modified.

Page S-19~S21, Figure S20. According to the **Comment (7)** of **Reviewer #1**, the adsorption structures of O, OH, O₂, and OOH on pristine MoS₂ and Ln-MoS₂ and the corresponding differential electronic densities induced by these adsorbates have been added.

Page S-22~S23, Figure S22. According to the **Comment (8)** of **Reviewer #1**, the pDOS spectra of the active S sites in pristine MoS₂ and Ln-MoS₂ before and after the adsorption of O, OH, and OOH have been added.

Page S-24~S25, Figure S23. According to the **Comment (8)** of **Reviewer #1**, the -pCOHP spectra for the S-O bonds after O and OH adsorptions for pristine MoS₂ and Ln-MoS₂ have been added.

Page S-27~S28, Figure S25. According to the **Comment (7)** of **Reviewer #4**, the free-energy diagrams for the ORR on all the fifteen Ln-MoS₂ surfaces at both 0 and 1.23 V have been added.

Page S-29, 2nd paragraph. According to the **Comment (5)** of **Reviewer #3**, the sentences of “To examine the stability of Ln-MoS₂ in the electrochemical conditions ... the chemical potential of the state of H₂S + S vacancy.” have been added to prove the stability of Ln-MoS₂ against the formation of H₂S.

Page S-29, 3rd paragraph. According to the **Comment (8)** of **Reviewer #4**, the sentences of “The

simulated surface Pourbaix diagrams ... reflecting the same modulating trend in surface reactivity.” have been modified.

Page S-29, 3rd paragraph. According to the **Comment (5)** of **Reviewer #3**, the sentences of “Furthermore, $\mu(\text{H}_2\text{S} + \text{V}_\text{s})$ keeps large positive values ... in Ln–S bond strength (i.e., reverse to the trend in E_f of Ln dopants, Figure 1c in the main text).” have been added.

Page S-31, Figure S27. According to the **Comment (5)** of **Reviewer #3**, the chemical potentials of different surface states for all the fifteen Ln-MoS₂ systems have been added.

Page S-32, Figure S28. According to the **Comment (8)** of **Reviewer #4**, the variations of the $U_{\text{O}^*/\text{clean}}$ and U_{limit} with respect to Ln dopant type have been added.

Page S-32, Figure S29. According to the **Comment (5)** of **Reviewer #3**, the $\mu(\text{V}_\text{s} + \text{H}_2\text{S})$ for all the fifteen Ln-MoS₂ surfaces at 0 V_{RHE} have been added.

Page S-33, Figure S30. According to the **Comment (6)** of **Reviewer #3**, the atomic structures for the initial, transition, and final states in the reaction paths of various ORR steps have been added.

Page S-34, 1st paragraph. According to the **Comment (6)** of **Reviewer #3**, the sentence of “and the atomic structures for different reaction paths are shown in Figure S30.” has been added.

Page S-40, Figure S36. According to the **Comment (11)** of **Reviewer #4**, the caption and the title of y axis have been modified.

Page S-41~S-42, 1st and 2nd paragraphs, Figure S37 and S38. According to the **Comment (8)** of **Reviewer #3**, the discussion about the thermal effect on current-potential polarization curves has been added.

Page S-43, Table S3. According to the **Comment (12)** of **Reviewer #4**, the test “metal surfaces” is used instead of “catalysts”.

Page S-44~S47, Section L. According to the **Comments (2), (3), and (10)** of **Reviewer #1**, the section of Comprehensive Numerical Tests towards Accurate DFT Calculations has been largely modified, with massive additional testing calculations have been added.

(A) Author replies to the comments by Reviewer #1

Comment (1):

In the work entitled “Oxygen Reduction Reaction on MoS₂: Activity Enhancement and Biperiodic Chemical Trends Induced by Lanthanide Dopant”, Hao et al. performed density functional theory (DFT) calculations and polarization curve simulations of oxygen reduction reaction (ORR) processes on 15 Ln-MoS₂ surfaces. They found not only that the ORR activity on MoS₂ surfaces is greatly enhanced by Ln doping, but also that the ORR activity varies with Ln type in a biperiodic chemical trend. The biperiodic chemical trend was also observed in the electronic structure, thermodynamic and kinetic quantities, such as dopant stability, dopant charge state, ORR-intermediate adsorption strength, reaction free energy (and U_{limit}) of ORR intermediates, characteristic potentials of polarization curves (U_{half} and U_{onset}), and current density. This is a very comprehensive theoretical study, and an interesting topic. Hopefully the author would publish it after solving the problems.

Reply (1):

Many thanks to this reviewer for the quite detailed review and specialized comments, as well as the nice summary and positive assessment on our work.

Comment (2):

Typically, DFT is not capable to describe Ln elements accurately, it is recommended to provide the reliability of their predications.

Reply (2):

This is a really nice and important point indeed deserving a detailed discussion and analysis in our Manuscript and Supplementary Information (SI), which we believe will also be highly useful for numerous related studies in the future. Apart from the supporting evidences in previous version of manuscript and SI, we have additionally carried out more testing calculations.

The computational reliability of this work can be answered from both technical and physical perspectives, which can be further divided into eight aspects below:

- a) **Negligible effect of self-interaction problem of localized 4f electrons.** The self-interaction problem of semilocal GGA functionals (e.g., PBE functional used here) should be the first possible issue people may wonder when dealing with localized orbitals (e.g., 4f here). It can be well tested using the *DFT plus Hubbard U* method, and the calculation results are listed in **Table S4** and discussed in the 1st paragraph of **Section L** in page 44 of supplementary information. Using the effective Hubbard U_{eff} parameter ranging from 0 to 6 eV (usual U_{eff}

for 4f element: 4~5 eV), we calculate the adsorption free energies of both O and OH on the active S sites in two representative systems, i.e., Ce-MoS₂ and Gd-MoS₂. We find that these adsorption free energies just vary by 0.00~0.03 eV with different values of U_{eff} . This negligible effect of U_{eff} is understandable, because the ORR intermediates are adsorbed on the S atom (with 2s and 2p valence electrons), and the self-interaction correction on the Ln dopants does not have sufficient indirect effect on the S—adsorbate bond strength.

- b) **Negligible effect of spin-orbit coupling (SOC).** Similar as the self-interaction problem, the effect of SOC also deserves an investigation for materials with localized orbitals (e.g., 4f here). According to our testing calculations, we find that the SOC changes the adsorption free energies of O and OH on Ce-MoS₂ and Gd-MoS₂ only by a very small amount of < 0.01 eV. The testing results are listed in **Table S5** and discussed in the 1st paragraph of **Section L** in SI (page S44-S45). The reason for the negligible SOC effect is the same as the negligible effect of self-interaction correction above, i.e., the surface reactivity is determined by the strength of S—adsorbate bond that has very weak SOC. The Ln dopant just has an indirect effect through modulating the Ln—S bond strength.
- c) **Reliable atomic structural model for Ln-MoS₂.** The structural model has been strictly validated from various perspectives in this work. First, we use DFT calculations to screen many possible doping sites to accommodate the Ln dopants, and find the most stable doping site at Mo site (see **Figure 1a** and **Figure S1**), which is closely consistent with the experimental observation using scanning transmission electron microscopy (STEM, see **Figure 1b**). Second, we calculate the phonon densities of states (DOS) of all the fifteen Ln-MoS₂ structures (**Figure S2**), where the absence of imaginary modes can clearly prove the dynamical stability of the used structures of Ln-MoS₂. Third, we calculate and analyze the Raman spectra of pristine MoS₂, La-MoS₂, Er-MoS₂, and Lu-MoS₂ (see **Figure S3** and **S4**, as well as the discussion in **Section B** in SI), and the observed red shifts of the Raman-active modes are closely consistent with the experimental observations, which can also validate the considered Ln-MoS₂ structures.
- d) **Validated supercell size for Ln-MoS₂.** The atomic Ln dopants in the experimental sample (see **Figure 1b**) are separated by a distance of ~20 Å, and the inter-dopant distance is ~11 Å in the 4×2√3 rectangular supercell used in this work (see the METHODS section in main text). To validate this supercell model, we test the supercell-size effect on the formation energies of Ln dopants (**Figure S39-S40**), adsorption free energies of O and OH (**Table S7**), and inter-dopant magnetic coupling (**Table S8**), all of which clearly reveal that the interaction between Ln dopants are negligibly weak, and the interaction energy has been stringently converged in the used 4×2√3 supercell.
- e) **Carefully tested reciprocal-point mesh and cutoff energy.** Different reciprocal-point meshes

and cutoff energies are used to calculate the adsorption free energies of O and OH on Ce-MoS₂ and Gd-MoS₂, and the energies can stringently converge within 0.02 eV (see **Table S6**) under the computational parameters used in this work (2×2×1 and 450 eV).

- f) **Strict statistical treatment of the atomic structures of water.** The water environment at finite temperatures has the dynamically random H₂O-molecule configurations. A thick enough water film is considered in this work, and the atomic structures are simulated using the ab-initio molecular dynamics method with 45000 steps at 300 K. After that, we select sufficient number of water-film structures to calculate the ORR reactions at the Ln-MoS₂/water interfaces. This strict theoretical treatment of water environment has facilitated us to quantitatively disentangle the roles of electric field, hydrogen bond, and water-molecule reorientation in the energetic effect of water film on ORR reactions. This is a key technical part for this work, thus we specially discuss the related results, mechanisms, and comparison with reported results for Pt(111)/water interfaces in the **Section B.1** (page 5-6), as well as in the **Section G of SI** (page S-13 ~ S-16).
- g) **Reliable current-potential polarization curves simulated in this work.** The theoretical accuracy on this point has been well supported by three facts. First, as described in the above points, the strict technical approach used for the kinetic simulation, including the treatment of water environment, constructed microkinetic model, and comprehensively tested computational parameters, can well guarantee the accuracy of obtained results. Second, although the polarization curves for the Ln-MoS₂ surfaces with activated ORR are first reported by this work, the simulated polarization curves have very similar variation trends with the available experimental curves for the activated ORR on other doped MoS₂ systems (P-MoS₂ and O-MoS₂, see **Figure 4c**), which is a clear evidence strongly indicating the accuracy of our theoretical approach to calculate the polarization curves for activated ORR processes. Third, the obtained current densities for Ln-MoS₂ at the standard potential of 0.9 V well comply with the famous volcano trend that has been established based on a large amount of existing experimental/theoretical results for ORR catalysts (see **Figure 5a**), which is another strong evidence supporting the accuracy of our theoretical results.
- h) **Generic biperiodic chemical trends with wide experimental evidences and unifying mechanism rationale.** In this work, the biperiodic chemical trends have been simultaneously observed in various properties, e.g., formation energies of Ln dopants (E_f , **Figure 1c**), adsorption free energies of different adsorbates on Ln-MoS₂ surfaces (ΔG_{ads} , **Figure 2e-f**, and **Figure S18**), limiting potential (U_{limit} , **Figure 3b**), half-wave potential (U_{half} , **Figure 4d**), onset potential (U_{onset} , **Figure S35**), and current density (j , **Figure 5b**), as well as some newly added calculation results in the new version of SI, i.e., the equilibrium potential between O* and clean surface ($U_{O^*/clean}$, **Figure S28**) and the chemical potential of reaction for the formation

of H₂S from Ln-MoS₂ ($\mu(\text{V}_s+\text{H}_2\text{S})$), **Figure S29**). Such generic appearance of the biperiodic trend in these totally different properties is undoubtedly not coincident, and has also been mutually correlated by a fundamental orbital-chemistry mechanism (i.e., intraatomic orbital hybridization plus interatomic bonding). Furthermore, many previous experimental studies have also reported similar biperiodic trends for different properties of Ln elements in different states, e.g., the sublimation heats of elemental Ln metals (**Figure S5a**), ionization potentials of Ln atoms (**Figure S5b**), homolytic Ln-F bond energies of LnF₃ molecules (**Figure S6**), and electronic transition energies in Ln atoms (**Figure S7d**). These experimental phenomena have also been well explained by the orbital-chemistry mechanism established in this work for the dopant stability and ORR activity of Ln-MoS₂. The success of this work in observing the generic chemical trend and proposing a unifying orbital-chemistry mechanism should have also been benefited from the high accuracy and capability of our theoretical method.

Comment (3):

It was reported that spin coupling and magnetic coupling associates with the catalytic activity [J. Am. Chem. Soc. 2018, 140 (45), 15149-15152; Adv. Powder Mater. 2022, 1(3), 100031], authors might also consider such coupling.

Reply (3):

These points are really nice and important! Many thanks to this reviewer for this comment and providing highly useful references, which we have used into our new SI to support our discussion about the newly added results for magnetism-related things.

According to this review comment, we have added more testing calculation results for the effects of spin-orbit coupling (SOC) and inter-dopant magnetic coupling, as well as the necessary testing calculations for supercell-size effect (see **Section L of SI**, page S-44 ~ S-47). Before answering this comment, it should be noted that the atomic structure for Ln dopant in MoS₂ layer determined by our structural screening (**Figure 1a** and **Section A of SI**) is consistent with the experimental measurement using scanning transmission electron microscopy (**Figure 1b**). Then, we use an appropriate supercell of MoS₂ (i.e., 4×2×3 supercell) to accommodate any atomic Ln dopant. Sufficient testing calculations for the supercell-size effect (**Table S7 ~ S8** and **Figure S39 ~ S40**) clearly show that this supercell size can yield well converged formation energies of Ln dopants and adsorption free energies of adsorbates.

Then, the effect of SOC is tested, the results of which have been described in detail in the above Reply (2): The SOC effect changes the adsorption free energies only by a negligible amount of < 0.01 eV (**Table S5** and **Section L of SI**). This negligible SOC effect is understandable, because it is

the active S site for the adsorbate to bond with, and the SOC on the Ln dopant does not have an observable direct effect on the S—adsorbate bond.

To investigate the magnetic coupling between neighboring Ln dopants in MoS₂, four Ln-containing supercells (2×2√3, 3×2√3, 4×2√3, and 5×2√3) are doubled along the x direction, which let us obtain two Ln dopants in the extended supercells. Both antiferromagnetic (AFM) and ferromagnetic (FM) configurations between these two Ln dopants are considered, and are calculated to have nearly the same energies with each other (difference ≤ 0.007 eV, **Table S8**) for the later three supercells. Therefore, the 4×2√3 supercell is large enough to model the experimental doped MoS₂ sample, and the inter-dopant magnetic coupling is very weak and the spin polarization of each dopant can be treated individually (i.e., a paramagnetic state). We can also say that it is the very weak superexchange interaction between neighboring Ln dopants that leads to a superparamagnetic coupling configuration.

In addition, it should be noted again that the ORR intermediates are adsorbed on the active S atoms, not on the Ln dopants here. This situation is different from that of transition-metal doped nitrogenated graphene [Yu, *Adv. Powder Mater.* 1, 100031 (2022); Li, *J. Am. Chem. Soc.* 140, 15149 (2018)], where the active sites exactly reside on the transition-metal dopants and their magnetic coupling effect may play a role in ORR activity. We have added some related discussion in **page S-45 ~ S-46 of SI**.

Comment (4):

The authors' definition of the limiting potential as $U_{\text{limit}} = 1.23 - \Delta G_{\text{max}}/e$ is different from the definition in many literature, one example is [ACS Appl. Mater. Interfaces 2020, 12, 52549-52559]. Authors are suggested to carefully check it.

Reply (4):

We have checked the related formula in our manuscript and the reference paper mentioned in this review comment [Zeng, *ACS Appl. Mater. Interfaces* 12, 52549 (2020)], both of which actually are correct. It should be noted first that overpotential (η) and limiting potential (U_{limit}) are two different parameters reflecting the common ORR activity, and their specific expressions also depend on the used electrode potential.

In that reference paper, it is the ORR overpotential **defined at 0 V** ($\eta = \Delta G_{\text{max}}/e + 1.23$), which actually corresponds to the number of 0.98, 0.70, 0.75, and 1.99 shown in the free-energy diagram here (**Figure 3a**). It indicates that a potential of $< 1.23 - \eta$ is needed to make all the ORR steps exothermic, i.e., this $1.23 - \eta$ equals the U_{limit} defined in this work. It should be noted that the U_{limit} here is defined at 1.23 V ($U_{\text{limit}} = 1.23 - \Delta G_{\text{max}}/e$) and can directly give out the highest potential value to make all the ORR steps exothermic.

To make the description clearer, we have modified the related sentences in Manuscript (**page 9, 2nd paragraph, line 3-8**): *“In a FED at 1.23 V, the free-energy change of the representative Sm-MoS₂ surface.”* Furthermore, to make the calculated U_{limit} values clearer and more comprehensive, we have also added the free-energy diagrams for the ORR processes at both 0 and 1.23 V for all the fifteen Ln-MoS₂ systems in **Figure S25 of SI** (section I, page S-27 ~ S-28).

Comment (5):

When evaluating the stability, the text says that the phonon spectrum was calculated “In addition, the calculated phonon spectra for Ln-MoS₂ (Figure S2) further prove the favorable dynamical stability of Ln-MoS₂.”, but Figure S2 presents the phonon density of states instead of the phonon spectrum. Authors only provide three systems in Figure S2, instead, all the considered systems are suggested to be given.

Reply (5):

The term “phonon spectrum” is a somewhat open concept that can indicate the dispersion of vibrational frequency, and in the previous manuscript, we used “phonon spectra” for phonon densities of states. In the new version of manuscript, we have explicitly used “phonon densities of states”. Furthermore, as suggested by this reviewer, we have calculated the phonon densities of states for all the fifteen Ln-MoS₂ systems, which are shown in **Figure S2 in SI** (page S-3). A related sentence in Manuscript (page 3, 1st paragraph, last 2 lines) is modified correspondingly: *“In addition, the calculated phonon densities of states (Figure S2) further prove the favorable dynamical stability of all the fifteen Ln-MoS₂ systems.”*

Comment (6):

Figure S20 shows that the potential limiting step for all the 15 catalysts is the first electron step, and from the equation of limiting potential versus free energy, it is clear that the limiting potential is inversely related to ΔG_{OOH} . Most of the catalysts in FIG. 3b correspond well with FIG. 2f, except for Pm, Sm, and Eu, please check the data.

Reply (6):

Many thanks for the close inspection of our results! Around a peak plateau of the biperiodic curve, it is possible for the influence of some secondary factors to show up. This is the reason why the calculated limiting potential and $\Delta G_{\text{ads}}(\text{OOH})$ curves do not have an ideal corresponding relationship with each other for Pm, Sm, and Eu.

The limiting potential is calculated based on the first electron transfer step, i.e., $\text{O}_2^* \rightarrow \text{OOH}^*$. Thus, U_{limit} depends on both the adsorption free energies of OOH and O₂. OOH* forms strong covalent bonding with Ln-MoS₂ surface, but O* much weaker physical+electrostatic bonding. Thus,

biperiodic trend is mainly determined by $\Delta G_{\text{ads}}(\text{OOH})$, and the minor influence of $\Delta G_{\text{ads}}(\text{O}_2)$ may show up at the peak plateaus, e.g., at Pm, Sm, and Eu.

To make our words more precise, we have modified a related sentence in Manuscript (page 9, 2nd paragraph, line 13-15): *“The U_{limits} for all the Ln-MoS₂ surfaces under water are shown in Figure 3b, where it can be seen that the magnitude is modulated between 0.31 and 0.54 V by a biperiodic chemical trend, and is almost inverse to the trend in $\Delta G_{\text{ads}}(\text{OOH})$ (Figure 2f).”*

Comment (7):

The authors compared only the Sm-MoS₂ structure with the pristine MoS₂ in Figure S16, and derived the difference between the whole Ln-doped systems and MoS₂, which is not rational, please complete the calculations for other doped systems.

Reply (7):

We have comprehensively carried out the related calculations suggested by this reviewer, please check the added differential electronic densities for O*, OH*, O₂*, and OOH* on all the 15 Ln-MoS₂ surfaces in the new version of SI (**Figure S20**, page S-19 ~ S-21). Having these complete set of data, one of our previous conclusions can be further emphasized, i.e., the Sm-MoS₂ system indeed is a nice representative surface for other Ln-MoS₂ systems, and the observed Ln-doping effects on bond length and electronic structure are quite similar.

Comment (8):

Similarly, FIG. 2h and i are calculated only for Sm-MoS₂ for comparison with MoS₂, which could not indicate universalism, please calculate the other 14 structures or remove the discussion of the other 14 structures in the ORR section.

Reply (8):

The corresponding pDOS and -pCOHP spectra for pristine MoS₂ and all the fifteen Ln-MoS₂ systems are added in the new version of SI (see **Figure S22** and **S23**, page S-22 ~ S-25). It not only gives out a complete set of evidences for the defect-pairing mechanism, but also further emphasizes our previous conclusion, i.e., the Sm-MoS₂ system indeed is a nice representative surface for other Ln-MoS₂ systems.

Many thanks for the above two comments, which have made our conclusions more strictly supported by comprehensive data sets.

Comment (9):

The authors say that FIG. 2 is the average of all Ln-doped structures, authors might give the

reason for such comparison.

Reply (9):

There are two reasons for the choice of averaged adsorption free energies to show in Figure 2g. First, we want to use a summarizing way to present the effects of both Ln doping and water environment, which can succinctly reveal the key mechanism and better motivate more related experimental/theoretical studies in the future.

Second, this summarizing presentation way actually has a very solid objective ground (i.e., the behaviors of adsorption free energies). To prove this, we have added the changes in ΔG_{ads} by both Ln doping and water environment for all the fifteen Ln-MoS₂ systems in the new version of SI (**Figure S19**, page S-17, namely the two panels below). From the new Figure S19, we can clearly see that the relative changes of $\Delta G_{\text{ads}S}$ ($\Delta\Delta G_{\text{ads}S}$) between different adsorbates (O, OH, O₂, OOH) are quite uniform for different Ln-MoS₂ systems. It is such uniform behavior of relative $\Delta\Delta G_{\text{ads}}$ data across all the fifteen Ln elements that makes the averaged adsorption free energies in Figure 2g a nice way to reveal the chemical effects of Ln doping and water environment.

Corresponding to the newly added Figure S19, we have also added more description in the **Section H of SI** (page S-17), and modified a related sentence in the Manuscript (page 7, 2nd paragraph, line 3-5): *“All the fifteen Ln-MoS₂ surfaces are averaged for each data point in Figure 2g to reveal the general effects of both Ln doping and water environment on different adsorbates, and these effects for all the adsorbates on all kinds of Ln-MoS₂ surfaces are shown in Figure S19.”*

Figure S19. The effects of Ln doping and water environment on the $\Delta G_{\text{ads}S}$ for different adsorbates on all the fifteen Ln-MoS₂ systems.

Comment (10):

The energy cutoff of 450 eV and kpoint of 2×2×1 might be tested for accuracy.

Reply (10):

More necessary testing calculations on the cutoff energy and reciprocal-point mesh, as well as the supercell size, have been carried out. The corresponding new data have been added and discussed in the **Section L of SI** (page S-44 ~ S-47). All of these testing results clearly indicate the stringent accuracy of the computational parameters used in our DFT calculations.

In addition, we have also added a declaring sentence in the Manuscript (METHODS section, page 14, 2^{ed} paragraph, the last 3 lines): *“More testing calculations on the spin-orbit coupling effect, cutoff energy, reciprocal-point mesh size, supercell size, and magnetic configurations are comprehensively carried out (see section L of SI), which further stringently validate the structural model and computational parameters considered in this work.”*

Comment (11):

It is known that good conductivity benefits the electrochemical process, and MoS₂ monolayer is semiconductor, authors might exploit the band structures of the Ln-doped MoS₂.

Reply (11):

In recent ten years, we have also been witnessing many publications reporting the promising applications of (conductive) defective MoS₂ systems in the fields of quantum transport and optics. Some nice recent publications (plus many references therein) can help us fully understand the importance of defective MoS₂ systems:

- [1] Chakraborty, *“Voltage-controlled quantum light from an atomically thin semiconductor”*, **Nature Nanotechnology** 10, 507–511 (2015).
- [2] He, *“Single quantum emitters in monolayer semiconductors”*, **Nature Nanotechnology** 10, 497–502 (2015).
- [3] Li, *“Approaching the quantum limit in two-dimensional semiconductor contacts”*, **Nature** 613, 274 (2023).

Although an in-depth analysis and exploitation of band structures is not necessary in this work focusing on surface electrochemistry, more information about the doping effect on the electronic structure should be helpful for motivating/guiding many related studies in the future, including our future studies on many possible behaviors and properties of MoS₂-based defects and interfaces under various environmental conditions. Therefore, apart from the newly added comprehensive electronic densities of states for all the fifteen Ln-MoS₂ systems in SI (**Figure S22**, page S-22 ~ S-23), we have also added two sentences in **page 8~9** in the Manuscript (the last line in page 8, and the beginning three lines in page 9), where a supporting experimental evidence [Li, **Nanomaterials**, 11, 769 (2021)] is also given: *“It is indispensable to have a conductive surface to freely exchange electrons during an ORR process. As seen from the pDOSs of Ln-MoS₂ (Figure 2h and Figure S22), the defect states at the Fermi level brought by Ln doping indeed will result in the p-type conductivity of MoS₂, which has also been proved by the field-effect experiment on Sm-MoS₂ [11].”*

Comment (12):

As for the water effect, what is the distance between MoS₂ and water layer?

Reply (12):

The distance between MoS₂ surface and water layer is defined to be the difference in vertical z coordinate between the H atom in the H₂O molecule closest to the surface S layer and the average z coordinate of all the surface S atoms.

The distributions of surface-water distances, as calculated from two groups of ab-initio molecular dynamics simulations, for the the representative Ce-MoS₂ system are newly added in **Figure S11** of SI (page S-13, also see the figure below), which is referred in the 1st paragraph of Section B.1 of Manuscript (page 5): “... the distance between water film and Ln-MoS₂ surface is around 2.1 Å (Figure S11)”, as well as in the METHODS section of Manuscript (page 15, line 1~2): “The weak interfacial interaction can be well proved by the relatively large interface distance (about 2.1 Å) in the simulated structures (Figure S11)”. Finally, the average distance is obtained from these distribution analyses, as shown by the figures below.

Figure S11. The interface distance between Ce-MoS₂ surface and water layer, which is defined as the vertical-coordinate difference between the H atom in the H₂O molecule closest to the Ce-MoS₂ surface and the surface S atom. The normal-distribution analyses on the calculated interface distances are also made.

Comment (13):

“valent electrons” should be “valence electrons”?

Reply (13):

The “valent” has been corrected to be “valence” all through the paper.

(B) Author replies to the comments by Reviewer #2

Comment (1):

The biperiodic chemical trend of lanthanide is known to us and has been thoroughly studied before, and the creativity of this work is yet to be further discovered. I can't recommend this manuscript to be published. The biperiodic chemical trends of lanthanide has long been known, which have been demonstrate in previous experiments and calculations (Brewer, L., *Systematics and the Properties of the Lanthanides*. First ed.; D. Reidel Publishing Company: 1982; Phys. Chem. Chem. Phys., 2013,15, 7839-7847; *Thermochimica Acta*,1992, 209, 175-188).

Reply (1):

It seems that our story and part of the mentioned references have been incorrectly interpreted by this comment.

First, as the title already clearly indicated, the novel contribution of our work is footed on the enhanced ORR activity with biperiodic chemical trend for lanthanide-doped MoS₂, for which we have proposed novel+generic orbital-chemistry and defect-pairing mechanisms based on our analyses on various electronic, thermodynamic, and kinetic properties. Considering the high degree of importance of (doped) MoS₂ in the scientific community in recent years, our fully novel discovery of ORR behaviors on Ln-MoS₂, and the strictly established mechanisms with high degree of generality, we believe our manuscript should be competitive enough to deserve a publication in Nature Communications.

Second, we have collected a large amount of experimental results (e.g., dopant structure, Raman spectra, various properties of Ln elements in different states with biperiodic trends, current-potential polarization curves of MoS₂-based materials, and current densities of many transition metals) for the comprehensive comparison, which can well support the high degrees of accuracy and generality of our theoretical work from different perspectives, without any negative influence on the novelty of our study for ORR on Ln-MoS₂. After a thorough literature investigation, people can also readily know the novelty of our orbital-chemistry mechanism for lanthanide-related studies, not to mention its unifying power in explaining all the biperiodic trends observed in various electronic, thermodynamic, and kinetic properties for different Ln-containing states (e.g., atoms, metals, compounds, and dopants).

The first reference mentioned above in this review comment [Brewer, *Systematics and the Properties of the Lanthanides*. 1982] actually is the reference [10] in our SI, and from it, we collected the experimentally-measured relative energies between different orbital-occupation configurations of Ln atoms (see **Figure S7d** in SI, page S-8). Such experimental data for Ln atoms are very nice evidences that can well support the generality of our orbital-chemistry mechanism,

but should not have any negative influence on the novelty of our work for ORR activity of Ln-MoS₂.

The second reference mentioned in this review comment [Xu, **Phys. Chem. Chem. Phys.** 15, 7839 (2013)] gives out the biperiodic chemical trend in homolytic Ln–F bond energies of LnF₃ molecules. This additionally well support the generality of the orbital-chemistry mechanism established in this work, and these energy data for LnF₃ molecules have been added into the new version of SI (see **Figure S6**, page S-7, also see below) and referred to in the Manuscript (page 4, 2nd paragraph, last sentence).

Figure S6. The homolytic Ln–F bond energies of LnF₃ molecules ($1/3\text{LnF}_3 \rightarrow 1/3\text{Ln} + \text{F}$) [8], among which the data for La, Ce, Pr, Nd, Eu, Gd, Dy, Ho, Er, and Tm are deduced from experimental measurements, and those for Pm, Sm, Tb, Yb, and Lu are obtained from quantum-chemistry calculations.

In the third reference mentioned in this review comment [Muraishi, **Thermochimica Acta** 209, 175 (1992)], the author just mentioned the word “double periodicity” in Abstract. However, the kinetic data therein for the thermal dehydration/decomposition of Lanthanide compounds do not exhibit any clear biperiodic trend, but are just sometimes separated into two groups at Eu~Gd, which can be proved by the plotted figures below for those experimental data.

The pre-exponential factor (A) and activation energy (E_a) for the thermal dehydration reactions of lanthanide oxalates, malonates, and succinates.

The pre-exponential factor (A) and activation energy (E_a) for the thermal decomposition reactions of lanthanide oxalates, malonates, and succinates.

Comment (2):

Layered MoS₂ is not the mainstream candidate for ORR with a fairly inert basal plane. The significance of choosing MoS₂ as the model needs further clarification.

Reply (2):

First, only the pristine MoS₂ surface has the inert basal plane, and the activated ORR can be found on the defect sites of MoS₂, which has been clearly described in our Introduction section (page 2, 2nd paragraph).

Second, it is the ORR on lanthanide-doped MoS₂ studied in this work, and its importance in the fields of electrocatalysis, optoelectronics, and solid-lubricating coatings has been described in the first three paragraphs of Introduction section (page 1~2). We can make another summary below:

- Promising for high-performance electrocatalysts:** Developing the non-noble metal catalysts is an important pursuit of the catalysis community, for which promoting the catalytic activity of MoS₂ basal plane is a promising approach [Jayabal, *J. Mater. Chem. A* 5, 24540 (2017); Zhu, *Chem. Soc. Rev.* 47, 4332 (2018); Zhang, *Chem. Soc. Rev.* 50, 9817 (2021)]. Due to the profound physical/chemical effects of 4f orbitals on the electronic structure of MoS₂, Ln dopants can bring attractive and remarkable change to the electrocatalytic activity of MoS₂.
- Guiding the protection of superior functional materials:** Our **Reply (11) to the Reviewer #1** has already pointed out the importance of MoS₂ in recent explorations of quantum-transport/optical nanodevices, which can be well reflected by the three recent publications (and the references therein) mentioned there [Chakraborty, *Nature Nanotechnology* 10, 507 (2015); He, *Nature Nanotechnology* 10, 497 (2015); Li, *Nature* 613, 274 (2023)]. Due to the profound effects of 4f orbitals, Ln-MoS₂ systems have superior optical properties, which is a

research focus in recent years [Bai, *Adv. Mater.* 28, 7472 (2016); Maddi, *Adv. Optical Mater.* 7, 1900753 (2019); Xu, *Angew. Chem. Int. Ed.* 57, 755 (2018)]. Furthermore, Ln and MoS₂ are also key materials for solid lubrication [Ye, *Surf. Coat. Technol.* 203, 1121 (2009)], and Ln-MoS₂ may become an important research object for the field of lubricating coatings, including the friction of related optoelectronic nanodevices. When working in realistic humid-atmosphere (or aqueous) environments, the ORR on Ln-MoS₂-based functional nanodevices and coatings is a critical and ubiquitous electrochemical process that can cause severe galvanic corrosion. This can be well exemplified by the corrosion phenomena on graphene coating supported by Cu [Prasai, *ACS Nano* 6, 1102 (2012)]. MoS₂ nanoparticles have the activated ORR, due to the high ratio of edges [Wang, *Chem. Eur. J.* 19, 11939 (2013); Hao, *ACS Appl. Mater. Interfaces* 11, 46327 (2019)], and the deposited MoS₂ nanoparticles can accelerate the corrosion of steel substrate [Büttgen, *Mater. Corros.* 49, 505 (1998)]. Therefore, for the long-term applications of Ln-MoS₂-based functional nanodevices and coatings in realistic environments, it is highly desired to comprehensively and deeply understand their ORR behaviors, which can well guide their appropriate protection against any detrimental corrosion.

Comment (3):

The particularity of Sm needs to be elucidated, why was it chosen as the ORR model?

Reply (3):

It should be more accurate to say that we have comprehensively calculated the electronic, thermodynamic, and kinetic properties for all the fifteen Ln-MoS₂ systems with various surface adsorption states, but just have a particular arrangement of the Figures in Manuscript and SI. There are a huge amount of data and plotted figures (> 300 panels in total), thus we need to choose a succinct way to facilitate our discussion on the ORR behaviors and mechanisms.

Therefore, we sometimes show the results for all the fifteen Ln-MoS₂ systems in the Manuscript (see **Figure 1d-e**, **Figure 2e-g**, **Figure 3b**, **Figure 4b-d**, **Figure 5a-b**), but sometimes will only give out the results of representative systems (e.g., Sm-MoS₂ and Ce-MoS₂, see **Figure 2b-d,h-l**, **Figure 3a,c**, **Figure 4a,e-f**) in the Manuscript, with the corresponding results for other Ln-MoS₂ systems shown in SI (see **Figure S2**, **Figure S8**, **Figure S19**, **Figure S20**, **Figure S22**, **Figure S23**, **Figure S25**, **Figure S26**, **Figure S27**, **Figure S34**).

In the Manuscript (the last sentence of Section II.A, page 5), we have already pointed out the reason why the representative systems are frequently used:

“It is also the similarity in Ln–S bonding character for all Ln-MoS₂ systems that allows us to select Ce- and Sm-MoS₂ as representatives in the following clarifications of many calculated properties

and mechanisms.”

Actually, the fruitful analysis realized in this paper have also clearly proved the efficacy of our arrangement of data/figures in the Manuscript and SI.

Comment (4):

The stability of Ln-doped MoS₂ is estimated by the formation energy. Have the phonon spectra been calculated to determine their dynamical stability?

Reply (4):

Yes, the dynamical stability of Ln-MoS₂ structures indeed has been proved by phonon densities of states.

In the previous version of SI, the phonon densities of states for three representative Ln-MoS₂ systems (Ln = La, Eu, Lu) have already been provided in Figure S2, which were referred to in the previous version of Manuscript (page 3, 1st paragraph, the last sentence): *“In addition, the calculated phonon spectra for Ln-MoS₂ (Figure S2) further prove the favorable dynamical stability of Ln-MoS₂.”*

In the new version of SI, we have provided the phonon densities of states for all the fifteen Ln-MoS₂ in Figure S2, which are also briefly discussed in page S-2, and referred to in the new version of Manuscript (page 3, 1st paragraph, the last sentence): *“In addition, the calculated phonon densities of states (Figure S2) further prove the favorable dynamical stability of all the fifteen Ln-MoS₂ systems.”*

Comment (5):

What is the doping concentration of the lanthanide? Whether this is consistent with the experiment.

Reply (5):

In the previous version of manuscript and SI, the structural details have already been described/shown (see Figure 1a,b, the METHODS section).

To more clearly clarify the structural details, we have added more discussion about the theoretical model, experimental sample, and necessary testing calculations in the new version of SI (see the Section L of SI, page S-45 ~ S-47). There, we can see that although the inter-dopant distance (~11 Å) in our theoretical structure is smaller than that in experimental sample (~20 Å) [Bai, *Adv. Mater.* 28, 7472 (2016); Xu, *Angew. Chem. Int. Ed.* 57, 755 (2018); Maddi, *Adv. Optical Mater.* 7, 1900753 (2019)], stringent convergences have been achieved for the dopant stability and surface adsorption free energies by our theoretical model (see **Table S7~S8, Figure S39~S40**).

Comment (6):

What is the active site of Ln-doped MoS₂, lanthanide, S or Mo? Whether the biperiodic chemical trend is affected by the lanthanide content?

Reply (6):

The active site resides on the surface S atom, and the Ln dopant indirectly modulates the reactivity of surface S atom by tuning the strength of Ln–S bond. This kind of information should have been conveyed at many places in our Manuscript and SI (please try searching “active S”), as well as by many structure figures for adsorbed surfaces (i.e., with adsorbate-S bonds).

The biperiodic chemical trend will not be affected by the dopant concentration, at least within a reasonable concentration range. This can be clearly proved by our testing calculations for the supercell-size effects on dopant stability and surface adsorption free energies (see the Section L of SI, page S-45 ~ S-47). An example group of testing results for Ln dopant stability is shown below.

Figure S40. The biperiodic trend of E_f for Ln-MoS₂ with different supercell sizes.

Comment (7):

Now that the authors have obtained the material structure (Fig. 1b), it is encouraged that the catalytic activity to be tested experimentally.

Reply (7):

Motivating many related experimental studies in the future is also an eager of us during the progress of this theoretical work, which has pushed us to make our simulation approach as strict as possible and the comparison with available experimental evidences as close+comprehensive as possible.

As described in our **Reply (2) to the Reviewer #1** above, a large amount of experimental results for various properties have been closely compared and explained:

- The atomic structures and Raman spectra of Ln-MoS₂ [Bai, *Adv. Mater.* 28, 7472 (2016)], as shown in Figure 1b and discussed in the Section B of SI.
- Ionization potentials of Ln atoms and sublimation heats of elemental Ln metals [Haynes, *CRC Handbook of Chemistry and Physics* (2016)], as shown in Figure S5.

- c) Homolytic Ln–F bond energies of LnF₃ molecules [Xu, *Phys. Chem. Chem. Phys.* 15, 7839 (2013)], as shown in Figure S6.
- d) Relative energies between different orbital-occupation configurations of Ln atoms [Brewer, *Systematics and the Properties of the Lanthanides* (1982)], as shown in Figure S7d.
- e) Current-potential polarization curves measured on available P/O-MoS₂ systems with activated ORR [Huang, *J. Mater. Chem. A* 3, 16050 (2015); *Chem. Commun.* 51, 7903 (2015)], as shown in Figure 4c.
- f) Current densities of various conventional metallic surfaces at the standard potential of 0.9 V [Schmidt, *Electrochim. Acta* 47, 3765 (2002); Nørskov, *J. Phys. Chem. B* 108, 17886 (2004); Zhang, *Angew. Chem. Int. Ed.* 117, 2170 (2005); Stamenkovic, *Angew. Chem. Int. Ed.* 45, 2897 (2006); Stamenkovic, *Science* 315, 493 (2007); Blizanac, *J. Phys. Chem. B* 110, 4735 (2006); Shao, *Langmuir* 22, 10409 (2006); Nilekar, *Surf. Sci.* 602, L89 (2008); Greeley, *Nature Chem.* 1, 552 (2009); Viswanathan, *ACS Catal.* 2, 1654 (2012); Stephens, *Energy Environ. Sci.* 5, 6744 (2012); Tanaka, *Electrocatalysis* 5, 354 (2014)], as shown in Figure 5a.

The comprehensive and reliable theoretical results obtained from our strictly established simulation approach, as well as the interesting biperiodic chemical trend and clear/in-depth mechanisms, are expected capable of motivating/guiding numerous experimental studies in different fields, e.g., electrocatalysis, optoelectronic devices, and lubricating coatings, in the future. We also hope it can become a pivotal reference for many related first-principles calculations, not only on the surface properties, but also on the microscopic mechanisms (e.g., orbital-chemistry and defect-pairing mechanisms).

Comment (8):

In Fig. 1d, the curves of different Ln doped MoS₂ need to be differentiated.

Reply (8):

In the manuscript, all the curves are plotted together in Figure 1d, which can clearly reveal the common intraatomic orbital hybridization and interatomic bonding in all the fifteen Ln-MoS₂ systems. In the new version of SI, we have individually shown all the fifteen curves in **Figure S8**.

Comment (9):

There are some clerical errors in the article, for example, in page 4, this is not figure 1f, but figure 1e.

Reply (9):

It has been righted up in the new version of Manuscript. Many thanks for the detailed review!

(C) Author replies to the comments by Reviewer #3

Comment (1):

The authors present a computational investigation of lanthanide-doped MoS₂ (Ln-doped, where lanthanide atoms substitute for Mo atoms) as active sites for oxygen reduction reaction (ORR). The authors show a biperiodic trend in the energetics of this system (ranging from the energy of doping to the binding energy of adsorbates) and kinetics of ORR across the lanthanide series. Interestingly, OH and OOH bind rather strongly on the Lanthanide doped sites (i.e. on the S atoms bound to the dopant) and this is the primary reason for the enhanced activity of ORR on these sites relative to MoS₂. The paper is technically strong and this reviewer particularly likes that the electrochemical analysis was based on a kinetic model (rather than based on the computational hydrogen electrode assessments alone).

Reply (1):

Many thanks to this reviewer for this nice summary of our work and the positive assessment.

Comment (2):

The first question relates to the importance and impact of this work which cannot be readily assessed. The predicted activity of the Ln-doped catalysts seems high compared to Pt(111), so these materials do sound promising. Are there any experimental reports supporting such high activity? The authors do not seem to allude to any literature that corroborates this.

Reply (2):

This should be the first research report on the current-potential polarization curves of Ln-MoS₂, as well as the discovered enhanced ORR with biperiodic chemical trend. In this situation, when strictness of the method, reliability of the obtained results, and generality of the fundamental mechanisms can be solidly validated by substantial evidences, the high degrees of importance and novelty of our work will be quite creditable. Then, it can be expected to become a pivotal reference for many related future experimental/theoretical studies in different fields, e.g., electrocatalysis, optoelectronic nanodevices, and functional coatings.

Two clear evidences that can well validate the accuracy and reliability of our DFT calculations and microkinetic simulations can be readily seen in our manuscript. First, although there is still no experimental polarization curve for Ln-MoS₂, the simulated curves here have the similar potential dependence as the experimental curves for other doped MoS₂ systems (i.e., P/O-MoS₂). Second, in the volcano plot (**Figure 5a**), the polarization currents at the standard potential of 0.9 V for the fifteen Ln-MoS₂ all closely comply with the famous volcano trend that has been established based

on the classical results of many noble-metal surfaces.

It should be noted that many systems can have higher ORR current densities than Pt(111), e.g., Pt/Pd(111) and Pt₃X alloy surfaces in Figure 5a, and it should not be a surprise to see some Ln-MoS₂ residing at the volcano maxima. For realistic applications, a high current density is not the only determining factor, which is the reason why Pt surface currently should still be an optimal candidate for realistic ORR applications. Pt still has the superiority in both synthesis and service stability. In this work, we find Ln doping can activate the ORR on MoS₂ surface, and Ln-MoS₂ themselves are also found to exhibit preferred structural/chemical stability, because the Ln dopants are protected by the two sandwiching S layers. In addition to the systematical stability assessments in our previous version of paper, according to the Comment (5) below by this reviewer, we have additionally calculated the chemical potential of reaction for the formation of H₂S from Ln-MoS₂, and the positive values also well prove the favored stability against the release of H₂S. The edges of MoS₂ flake have much lower stability, and may release H₂S in acid conditions. For the future realistic application of Ln-MoS₂, there should be other technical issue to overcome, e.g., how to synthesize large-scale layer to decrease the ratio of edges, or how to passivate/protect the edges from contacting with environments. While, this should be fully out of the realm of this work, which focuses on the effect of atomic Ln dopants on ORR activity of MoS₂ basal plane.

In addition, as described in the above **Reply (2) to the Reviewer #1** and **Reply (7) to the Reviewer #2**, the strictness of our simulation approach and reliability of our results have been systematically proved by the close+comprehensive comparison with a large amount of available experimental data for different properties, as well as by establishing in-depth microscopic mechanisms (e.g., orbital-chemistry and defect-pairing mechanisms) that have a high degree of generality. The interesting biperiodic chemical trend and microscopic mechanisms are novel findings of this work, which can inspire many related studies in the future (e.g., transition-metal doped MoS₂, see our **Reply (4) below to this Reviewer**).

Comment (3):

The authors also mention that studying ORR is relevant to understand the mechanism of corrosion in such materials but it is unclear how big a challenge this is (i.e. is this problem the main impediment in deploying these materials). Further, if this issue is an important motivator, then is the chosen model (water-2D material interface) the correct description of the reactive environment?

Reply (3):

This is a good comment, which is partly related with some background knowledge. We can answer it from two aspects, i.e., the contemporary development trend of corrosion studies for

two-dimensional materials and the understanding of galvanic corrosion induced by ORR.

First, the importance of corrosion for doped MoS₂ can be well understood from the contemporary development trend of corrosion studies for two-dimensional materials. In the research field of two-dimensional materials, their various possible applications have been intensively explored in recent two decades. In reality, the application potential of a new material is not only determined by their properties to be exploited, but also limited by their structural/chemical stability against the attack of aggressive environmental agents (e.g., O₂ and H₂O). In regular environmental conditions (e.g., humid ambient atmospheres and aqueous solutions), the corrosion resistance is a key evaluation factor. Similar as the research-tide order between graphene, h-BN, and transition-metal dichalcogenides (TMD = MoS₂, WS₂, MoSe₂, WSe₂ ...), the corrosion studies for both graphene and h-BN started earlier than TMD, and have been keeping a hot topics in recent 10 years, which can be well proved by a lot of publications in many high-impact journals, e.g., Prasai, *ACS Nano* 6, 1102 (2012); Schriver, *ACS Nano* 7, 5763 (2013); Weatherup, *J. Am. Chem. Soc.* 137, 14358 (2015); Chilkoor, *ACS Nano* 14, 14809 (2020); Zhao, *Nano Lett.* 21, 1161 (2021); Luo, *Adv. Mater.* 33, 2102697 (2021); Wu, *Adv. Funct. Mater.* 32, 2110264 (2022). As the most important TMD system, MoS₂ also has various promising applications (e.g., in catalysis, optics, electronic transport, and quantum information), with an increasing research intensity in recent years, and it already has been applied as an important solid lubricant for a long term. Foreseen from the development of precedent corrosion studies of graphene and h-BN, we can naturally expect that the corrosion study will become a more and more urgent task for the MoS₂ systems in many important material situations and environmental conditions. The addition of 4f lanthanide dopants has been proved to bring profound effects (see Introduction section of our Manuscript), thus Ln-MoS₂ systems can endow us with the great chance to have a broader exploration of electrochemical properties and mechanisms. Apart from in the aqueous environments, in regular humid atmospheres, it is also an ultrathin water film usually formed on surface that has promoted the corrosion on graphene/metal and h-BN/metal systems, as reported in the above publications. Therefore, the 2D water films constructed in the structural models for our theoretical calculations are well footed on the realistic situations.

Second, corrosion is a process defined as the prolonged dissolution/oxidation of a material working in an environment. ORR is a critical and ubiquitous cathodic electrochemical process that can cause severe galvanic corrosion in regular oxic atmospheres and solutions, which has been summarized in our recent invited review paper [Hao, *Curr. Opin. Electrochem.* 34, 101008 (2022)]. To explain more in detail here: An ORR process happening on a stable 2D material/coating will consume electrons, which can make the surrounding materials (e.g., metal substrates and connecting wires) become positively charged and easier to be oxidized and corroded by environmental agents. The promoted corrosion of Cu substrate by defective graphene coating can

be well exemplified by Schriver's observation [*ACS Nano* 7, 5763 (2013), see the figure below], which is ascribed to the electron transfer from metal to graphene (namely a galvanic effect at the graphene/metal interface). In experiment, it has also been found that the MoS₂ edge has a high ORR activity [Wang, *Chem. Eur. J.* 19, 11939 (2013)], which can well explain the accelerated corrosion of the steel substrate deposited with MoS₂ nanoparticles [Büttgen, *Mater. Corros.* 49, 505 (1998)]. Considering this importance of ORR for corrosion, it definitely is highly meaningful to investigate its behavior and mechanism for the novel Ln-MoS₂ series, which will be important for the long-term applications of Ln-MoS₂-based functional nanodevices and coatings in realistic environments.

It should be noted that the general/basic background things related with this comment cannot be all described in detail in this research paper, which mainly focuses on the ORR behaviors and mechanisms of Ln-MoS₂. It is appropriate for us to have a brief summary of the related things in the Introduction section, and the interested readers can refer to the cited references for more systematic background knowledge.

The corrosion progresses on the surfaces of bare and graphene coated Cu in an ambient atmosphere (location: University of California at Berkeley) [Schriver, *ACS Nano* 7, 5763 (2013)].

Comment (4):

The biperiodic trend is the particularly interesting and intriguing result of this work. Is there any experimental evidence for such behavior across doped 2D materials, even if not Ln-MoS₂?

Reply (4):

First, according to our knowledge, it should be the first report about the biperiodic trend for ORR on Ln-MoS₂, which has endowed our story with a high degree of novelty.

Second, to make our theoretical results fully reliable, we have to perform stringent validations from both technical and physical perspectives. To achieve this purpose, as explained in our **Reply**

(2) to Reviewer #1 and Reply (1) to Reviewer #2, we have performed a huge amount of testing calculations and comprehensively explained the similar biperiodic trends in various properties of Ln elements in different states (e.g., atoms, elemental metals, compound molecules, and solid dopants). Furthermore, all of those biperiodic trends reported in both our paper and literature can be simultaneously unified by the established orbital-chemistry mechanism, which is also a proof for the technical and physical reliability of our work.

Third, we indeed have calculated many other doped MoS₂ systems, including the ones with transition-metal (TM) dopants. Although they cannot be included in this work, we would share some new findings with this reviewer here. Different from the highly localized 4*f* orbitals on Ln dopants that indirectly modulate the Ln–S bonding by changing the intraatomic orbital hybridization, the 3*d* orbitals at transition-metal dopants have more considerable contribution to the direct interatomic bonding. Thus, the chemical modulating trend for TM-MoS₂ partly behaves biperiodic-like, but partly mixes with the direct interatomic bonding effect of 3*d* orbitals. Inspired by this work and our previous studies on TM-based systems [Huang, *Acta Mater.* 113, 311 (2016); *npj Mater. Degrad.* 3, 26 (2019)], it should not be difficult to deeply reveal the core mechanisms for the ORR on many MoS₂-based systems other than Ln-MoS₂. Therefore, there should be a high possibility for this paper to become an important reference for many follow-up studies.

Comment (5):

Are these sites stable during electrocatalysis? For instance, what are the energetics of the loss of S atoms bound to the dopant? In general, substitutional dopants can also weaken the metal-sulfur bonds which can then make it easy to lose the adjacent S atoms electrochemically or thermochemically. What are the energetics of the competing reaction $2\text{H}^+ + 2\text{e}^- + \text{S} \rightarrow \text{H}_2\text{S} + \text{vac}??$ Understanding this might be key to knowing if these catalysts will be stable.

Reply (5):

This is a really nice suggestion! Many thanks to this reviewer for this comment that has helped us get more novel findings!

When a defect in MoS₂ has a low stability, the release of H₂S is suspected. Apart from the systematical stability validations (e.g., structural screening, formation energies, phonon spectra, and molecular-dynamics simulations) shown in our previous version of Manuscript and SI, as guided by this comment, we have additionally calculated the chemical potential of reaction (μ) for the release process of H₂S [surface + 2(H⁺ + e⁻) → V_S + H₂S_(g)], which yields very helpful and interesting results. This μ is defined to be the chemical potential of the V_S + H₂S_(g) state. In the new version of Manuscript and SI, the calculated $\mu(\text{H}_2\text{S} + \text{V}_\text{S})$ s are shown in **Figure 3d** (page 8), **Figure S27** (page S-31), and **Figure S29** (page S-32), where the high positive $\mu(\text{H}_2\text{S} + \text{V}_\text{S})$ values (+1.2 ~

+2.2 eV) indicate the preferred stability of Ln-MoS₂ against the loss of active S atom, and the biperiodic chemical trend also clearly exhibits in $\mu(\text{H}_2\text{S} + \text{V}_\text{S})$ (see **Figure S29** in SI). The later can be also readily explained by the orbital-chemistry mechanism established in this work, and mainly depends on the variation in the Ln–S bond strength.

Some necessary discussion on these new results has been added in the new Manuscript (**page 9, the last paragraph**): “In addition, the possible release of H₂S from defective sites of MoS₂ [41] is also considered in our electrochemical simulation here, where an active S atom (bonding with the Ln dopant) is extracted out to form a H₂S molecule, with a S vacancy (V_S) left behind. From the persistent large positive $\mu(\text{H}_2\text{S}+\text{V}_\text{S})$ s for all Ln-MoS₂ surfaces at 0 ~ 1.23 VRHE (Figure 3d and Figure S27), we can deduce that the ORR-active S atoms are stable against the formation of H₂S. It is interesting to observe that $\mu(\text{H}_2\text{S}+\text{V}_\text{S})$ also generally has a biperiodic chemical trend (Figure S29) reverse to that of E_f (Figure 1c), because the less strong a Ln–S bond is (higher in E_f), the less energy cost to form H₂S.”

Comment (6):

The current density of the catalyst is related to the active site density of Ln-doped MoS₂. In this context, what is the motivation for considering the chosen supercell dimension? Is it based on any synthesis information? What is the maximum expected extent of Ln doping on MoS₂? Relatedly, if indeed, more Ln doping is possible than even considered in the calculations, how close can two dopant atoms get? A more substantive discussion of the doping density is needed to justify the high predicted activity on these catalysts.

Reply (6):

This is related with the supercell-size effect, which has been explained in detail above in our **Reply (2) to Reviewer #1** (item d) and **Reply (5)+(6) to Reviewer #2**.

The atomic Ln dopants in the experimental sample (see Figure 1b) are separated by a distance of ~20 Å with a good dispersion [dopant concentration at 0.3~1.0%, Bai, *Adv. Mater.* 28, 7472 (2016); Xu, *Angew. Chem. Int. Ed.* 57, 755 (2018); Maddi, *Adv. Optical Mater.* 7, 1900753 (2019)]. The inter-dopant distance is ~11 Å in the 4×2×3 rectangular supercell considered here (dopant concentration at 2.1%, see the METHODS section in main text).

To validate our structural model, we have additionally performed comprehensively testing calculations for the supercell-size effect on the formation energies of Ln dopants (**Figure S39-S40**), adsorption free energies of O and OH (**Table S7**), and inter-dopant magnetic coupling (**Table S8**). All these results reveal that the inter-dopant interaction is negligibly weak, and the interaction energy has been stringently converged by the used 4×2×3 supercell, i.e., the monomer dopant state in the experimental samples has already been well modeled by the used supercell.

It can be seen from the E_f -distance relationship as shown in the new Figure S39 (also see below), there is some attractive interaction between neighboring Ln dopants (~ 0.3 eV, about 3% of E_f) when the distance gets short enough (e.g., ≤ 5 Å). However, a small Ln concentration should be preferred by experimental fabrication process (e.g., magnetron sputtering), because too high Ln content may make the two-dimensional lattice structure unable to be obtained. The determination of the upper-limit Ln content not only is a physical problem, but should be a matter largely dependent on the used experimental technique. In this work, we use the previously reported experimental samples as the reference to validate our theoretical structure model.

Figure S39. The E_f of Ce- and Gd-MoS₂ with different supercell sizes.

Comment (7):

As pointed out earlier, a strength of this work is the development of j-V plot computationally using a microkinetic model. The authors, however, have not discussed how exactly the barriers were calculated. From some images in the SI, it seems like the barriers are calculated using CI-NEB calculations of a proton transfer (or proton-coupled electron transfer) from a hydronium ion to the surface adsorbate (O, OH, OO, OOH). More details about these calculations are needed (e.g. how do they choose their initial states for the NEB, etc.)

Reply (7):

In the new version of SI (**Figure S30**, page S-33, also shown below), we have additionally shown the atomic structures of the initial, transition, and final states in the calculated CI-NEB paths for the reactions of all the possible ORR intermediates at the representative water/Sm-MoS₂ interface. In the new version of Manuscript (METHODS section, page 14, 1st paragraph), we have also added more related details: *“The protonation rate of a surface adsorbate is limited by the reaction at water/MoS₂ interface, because a proton can quickly reach the electrical double layer due to its very low diffusion barrier in water (0.07 ~ 0.11 eV [63]). To model this rate-limiting interfacial step, a H atom is placed on a water molecule nearby the adsorbate to form a H₃O unit, and the relaxed structural model is used as the initial state for the CI-NEB path (see Figure S30).”*

Figure S30. The atomic structures for the initial states (IS), transition states (TS), and final states (FS) in the reaction paths of various ORR steps on the representative Sm-MoS₂ surface under the optimal H-2O water configuration. The H atoms participating in the reactions are highlighted in light blue color.

Comment (8):

The microkinetic model is pretty informative about the rate-determining steps and abundant surface intermediates and their variation across lanthanides. This reviewer recommends comparing the surface coverage from the model with the surface Pourbaix diagrams. Ideally, the authors could consider building a kinetic phase diagram and comparing it with the thermodynamic phase diagram to understand how far the system may be away from equilibrium.

Reply (8):

In the surface coverage curves simulated from microkinetic model (**Figure 4f**), the critical potential (0.88 V) is for sure identical to the phase boundary in Pourbaix diagram (**Figure 3c**). The kinetic coverage curves (**Figure 4f**) can further reveal the continuous variation of surface coverage with potential, while the thermodynamic Pourbaix diagram only presents the abrupt change in the most stable surface state (**Figure 3c**).

In addition, as we already stated in our Manuscript, the thermodynamic and kinetic activities can be indicated by U_{limit} (**Figure 3b**) and U_{onset} (**Figure S35**). It can be seen that the former is lower by ~0.4 V than the later for Ln-MoS₂ systems, and we have added a sentence in the new version of Manuscript (page 11, 2nd paragraph): *“The U_{onsets} are higher than the U_{limits} by 0.45 ± 0.05 V, quantitatively indicating how far the kinetic activity is away from the thermodynamic threshold.”*

Comment (9):

It seems like, right now, the authors only consider reactions 1-6 in their microkinetic model. Why not consider all ten reactions given in the SI in the model and let it determine the flux-carrying pathways under various scenarios? Since some reactions are thermochemical (and all reactions are ultimately dependent on temperature), the model will also then enable an understanding of the effect of temperature on the ORR activity. This can be valuable to predict the impact of higher operating temperatures in an electrochemistry context or in the context of galvanic corrosion.

Reply (9):

Whether an ORR step deserves our consideration in the simultaneous equations of our microkinetic model is determined by its relative reaction rate (activation energy, E_a) with respect to other competing steps.

According to **Figure 4a** in the Manuscript, the activation energies (E_a s) of the two dissociative steps ($O_2^* \rightarrow 2O^*$ and $OOH^* \rightarrow O^* + OH^*$) are as high as ~ 1.4 and ~ 0.5 eV, while the E_a s of the two competing associative steps ($O_2^* \rightarrow OOH^*$ and $OOH^* \rightarrow O^* + H_2O$) are only ~ 0.2 and ~ 0.03 eV. It should be for sure that the latter two will have much higher reaction constants than the former two. To quantitatively prove this claim, the rate constants of these steps at both 25 and 100 °C are calculated by transition-state theory ($k = k_B T/h \times \exp(-E_a/k_B/T)$), and the results are shown in the Table below. The extremely large difference between these two groups of reaction paths ($>10^6$ times) shows that the two dissociative steps can totally be neglected in the current-density calculation.

The rate constants of two groups of competitive reactions at different temperature.

	$O_2^* \rightarrow 2O^*$	$O_2^* \rightarrow OOH^*$	$OOH^* \rightarrow O^* + OH^*$	$OOH^* \rightarrow O^* + H_2O$
25 °C	1.34E-11	2.59E+09	2.20E+04	1.93E+12
100 °C	9.60E-07	1.55E+10	1.37E+06	3.06E+12

We have added some related discussion in the new version of Manuscript (page 10, last paragraph; page 11, line 1): “It can be seen that the two dissociative steps of $O_2^* \rightarrow 2O^*$ and $OOH^* \rightarrow O^* + OH^*$ have E_a s (about 1.4 and 0.5 eV) much higher than those of the competing associative steps of $O_2^* \rightarrow OOH^*$ and $OOH^* \rightarrow O^* + H_2O$ (about 0.2 and 0.03 eV), respectively. Then, the reaction rates of the former two dissociative steps at room temperature will be lower than their respective counterpart associative steps by $\geq 10^7$ times, and can be excluded from the possible ORR pathway.”

The reviewer’s suggestion to calculate the temperature effect is pretty cool! We have predicted both the effects of electrode rotation speed (900 ~ 3200 rpm) and thermal effect (25 ~ 60 °C) on the polarization curve and U_{half} for Sm-MoS₂ (**Figure S37**, new version of SI). The factors

contributing to thermal effect are also disentangled in **Figure S38**, which are also discussed in page S-41~S42 of SI. These results are also referred to in the new version of Manuscript (page 11, 2nd paragraph): “More detailed information about each individual $j-U$ curve, the derived onset potentials (U_{onset}), adsorbate coverages, the effect of RDE rotation speed (from 900 to 3200 rpm), and thermal effect (from 25 to 60 °C) can be found in Figure S34-S38.”

Figure S37. The simulated polarization curves and corresponding half-wave potentials for Sm-MoS₂ at different rotation speeds (upper two panels) and temperatures (lower two panels).

(D) Author replies to the comments by Reviewer #4

Comment (1):

This is an interesting study which can be improved, see (minor and major) comments below.

Reply (1):

Thank you very much for this positive assessment for our work!

Comment (2):

The hyperlink to ref 19 (Büttgen et al.) is dead.

Reply (2):

The previous DOI of Ref. 19 was incorrect, which has been righted up.

Comment (3):

Why is the weighted sum of ionization potentials shown in Figure 1c, and where do the coefficients 1 and 0.2 for IP₃ and IP₄ respectively, come from?

Reply (3):

This is a good point, and such numerical choice is related with the origins of IP_3 and IP_4 . Both IP_3 and IP_4 are associated with the excitation of 4f electrons, but IP_1 and IP_2 are not. This can be clearly reflected by their chemical trends (**Figure S5b**, page S-6, also shown below), where the variation curves of IP_3 and IP_4 with Ln type exhibit the biperiodic trends. Furthermore, with more and more electrons excited out of Ln atoms, the unexpected effect of net positive charge on IP will increase correspondingly. Thus, if we want to make a linear summation of IP_3 and IP_4 to present an overall biperiodic trend, IP_3 should have a higher weight for an optimized numerical presentation. We have added a description in page S-4 of the new version of SI: *“Similar biperiodic trend is also observed in the third and fourth ionization potentials of Ln atoms (IP, Figure S5b) [7], both of which can reflect the ionization capacity of 4f electrons. We make a weighted linear summation of $IP_3+0.2\times IP_4$, where the coefficients are first obtained by roughly fitting to the formation energies of Ln dopants, and then optimized to be 0.2 for a good trend presentation. The overall biperiodic trend of $IP_3+0.2\times IP_4$ is compared with the variation trend in the formation energy of Ln-MoS₂ (see main text, Figure 1c).”*

Figure S5. (b) The ionization potentials (IP) of Ln atoms. We also make a weighted linear summation of IP ($IP_3 + 0.2 \times IP_4$) to compare with the variation trend in the formation energy of Ln-MoS₂ (see main text, Figure 1c).

Comment (4):

The surface potential part in Methods B is wrong. It is not RHE, but SHE that can be related to the absolute electrode potential of 4.44 V. Then RHE is obtained by including the pH dependence $0.059 \times \text{pH}$.

Reply (4):

Thanks a lot for careful reading and kind reminding! We have changed the potential reference of the surface potential to be the SHE. Please check the changes in **the caption of Figure 1e** and **METHODS section** in the new version of Manuscript.

Comment (5):

Even though it is stated in the Methods that RHE is used throughout the paper unless otherwise specified, this is at the very end. I suggest that it is mentioned the first time it is used in the text and referring to the Methods for the details.

Reply (5):

This is a good remind! We have defined the abbreviation for “RHE” in the new Manuscript (page 9, 2nd paragraph): “The reversible hydrogen electrode (RHE) is used as the default potential reference in this work unless otherwise specified”.

Comment (6):

In Section B.1., O₂, O, OH, and OOH are mentioned as examples of ORR intermediates. Isn't that all of them?

Reply (6):

To make our words more precise, the first sentence of Section B.1 has been modified to be: “An ORR process mainly consists of the adsorptions and transitions of O_2 , O , OH , and OOH intermediates”.

Comment (7):

I would have preferred to have the energy diagrams in SI section I drawn at 0 V vs. RHE. I struggled to see the relation between U_{limit} and the largest step (though admittedly I am more used to electrooxidation reactions)

Reply (7):

In the new version of SI (**Figure S25**, page S-27 ~ S-28), the free-energy diagrams at both 0 and 1.23 V have been provided for all the fifteen Ln-MoS₂ systems.

Comment (8):

In the SI, section J, there is only one degree of freedom for the Pourbaix diagrams on the RHE scale, so the data could be presented in a table with U_{limit} and $U_{O^*/\text{clean}}$ equilibrium.

Reply (8):

In the new version of SI (**Figure S28**, page S-32), both the values of U_{limit} and $U_{O^*/\text{clean}}$ have been summarized together, which can facilitate a straight comparison. Both the Pourbaix diagrams (**Figure S26**) and associated chemical potentials (**Figure S27**) for all the fifteen Ln-MoS₂ systems can be jointly studied to derive the complete electrochemical information.

Comment (9):

In the SI, section K, it says “It can be seen that the dissociation energy barrier of O_2^* is as high as 1.26 ~ 1.50 eV and then is difficult to occur at room temperature, ...”. I suggest a change to “...and then is difficult to overcome at room temperature, ...”. A bit further down, on page S-20, it says “The current density (j) can be deduced...”. I would have written “...can be calculated...”.

Reply (9):

Yes, it indeed is better. The modifications at these two places have been done.

Comment (10):

At the end of page 9 and in the figure text for Figure 4b, it says “liner” but should say “linear”. On page S-22, there are two occurrences of “liner”, one in the main text and one in the text of

Figure S24.

Reply (10):

Corrections have been done all through the manuscript and SI.

Comment (11):

In Figure S27, the label for the y axis is “Coverage (θ)” but the other y axis labels have the form Quantity (Unit), so I would suggest to either remove the parenthesis or write something like “fraction of ML”.

Reply (11):

We have removed “(θ)” in the **Figure 4f** and that Figure S27 (= **Figure S36** in the new SI).

Comment (12):

In Table S3, there is a capitalized “Catalysts” which should be “catalysts”.

Reply (12):

It has been changed to be “metal surfaces”.

Comment (13):

In SI, section L, it says “...S atom, for which the PBE-GGA functional is accurate...”. Could I have a source for that claim?

Reply (13):

The chemical S–O bonding associated with the adsorption of ORR intermediates only involves the regular $2s$ and $2p$ orbitals, which do not suffer some issues like electronic self-interaction. First, the DFT accuracy here has been well validated by comparing the j-U curves for Ln-MoS₂ with the available experimental curves for P- and O-doped MoS₂ that have similar active ORR (Figure 4c), as well as by the close comply with volcano trend (Figure 5a). Second, from the famous original paper of PBE-GGA [Perdew, **Phys. Rev. Lett.** 77, 3865 (1996)], even the atomization energies of many molecules with sp electrons have a mean absolute error of only 7.9 kcal/mol, the energetic error for the transition of these regular $2sp$ -type atoms between different bonding states should be much smaller. Third, this conclusion is also partly of empirical knowledge, which has also been proved by many times during our density-functional theory calculations in recent 10 years. For example, for various transition-metal (hydr)oxides, the stability order for different structures at a certain composition can be always accurately predicted by PBE-GGA, the same as other higher-

level functionals (i.e., metaGGA and hybrid functionals) [Huang, *npj Mater. Degrad.* 3, 26 (2019)]. Perdew's original paper for PBE-GGA has also been followed by numerous theoretical calculations in the quantum-chemistry field, where PBE-GGA functional has been frequently used as the calibration reference.

In the new version of SI (**Section L**, page S-44), we have also added some discussion about the successful application of PBE-GGA functional for the surface reactivity of defective MoS₂ systems, which have been verified by experiments, e.g., many transition-metal doped MoS₂ [Deng, *Energy Environ. Sci.* 8, 1594 (2015)], MoS₂ with vacancy and strain [Li, *Nat. Mater.* 15, 48 (2016)], V_s generation on MoS₂ [Tsai, *Nat. Commun.* 8, 15113 (2018)].

REVIEWERS' COMMENTS

Reviewer #1 (Remarks to the Author):

Authors well addressed all my concerns in the revised version, I recommend publication in the current form.

Reviewer #2 (Remarks to the Author):

I appreciate the efforts made by the authors in addressing the concerns. The authors have relatively adequately addressed the raised issues. Therefore, I would like to recommend its acceptance for publication.

Reviewer #3 (Remarks to the Author):

The authors have addressed the questions of this reviewer in detail. The importance of this particular class of doped MoS₂ is still unclear, however, this paper is technically very strong and the biperiodicity is quite interesting, and this work could encourage follow-on experimental analysis. I recommend publication without any reservations.

Author Response to the Reviewers

We would like to express our deep gratitude to all the reviewers for their great effort, precious time, and helpful comments. Please check the **list of changes** having been made in the new manuscript and **our point-by-point replies** in the following.

(A) List of Changes

(The changes are colored in **red** in the **Manuscript_with_changes_marked.pdf** file)

- (1) Page 1. The title has been modified according to the editorial suggestion in the AUTHOR CHECKLIST.
- (2) Page 1. There is an affiliation missing for the last author (Liang-Feng Huang), and it has been added as the third affiliation in both the manuscript and supplementary information.
- (3) Page 1. The Abstract has been modified according to the editorial suggestion in the AUTHOR CHECKLIST.
- (4) Page 2, 1st paragraph, the last 4 lines. As suggested by Reviewer #3, Some experimental measurements for lanthanide-doped MoS₂ have been additionally described to emphasize the importance of this group of materials.
- (5) All the titles for the subsections of Results and Discussion are modified according to the editorial suggestion in the AUTHOR CHECKLIST.
- (6) All the figures are moved to the end of the manuscript, and the figure items and captions are also modified according to the editorial requirements in the AUTHOR CHECKLIST.
- (7) The complete information for the references has been provided as required in the AUTHOR CHECKLIST.
- (8) Page 8, 2nd paragraph, line 4-11. According to the editorial suggestion in the AUTHOR CHECKLIST, a brief comparison or discussion on current literature has been made to further indicate the advancement of our simulation approach.
- (9) Many minor English problems have been corrected all through the manuscript (not marked in red).

(B) Author Response to the Reviewers

Comment by Reviewer #1: Authors well addressed all my concerns in the revised version, I recommend publication in the current form.

Author Reply: Many thanks to this reviewer for this final positive evaluation!

Comment by Reviewer #2: I appreciate the efforts made by the authors in addressing the concerns. The authors have relatively adequately addressed the raised issues. Therefore, I would like to recommend its acceptance for publication.

Author Reply: Many thanks to this reviewer for this final positive evaluation!

Comment by Reviewer #3: The authors have addressed the questions of this reviewer in detail. The importance of this particular class of doped MoS₂ is still unclear, however, this paper is technically very strong and the biperiodicity is quite interesting, and this work could encourage follow-on experimental analysis. I recommend publication without any reservations.

Author Reply: Many thanks to this reviewer for this final positive evaluation!

In addition to existing description for the importance of lanthanide-doped MoS₂ in the Introduction section, we have further emphasized the importance (in page 2, 1st paragraph, last 4 lines) by introducing their superior optoelectronic properties more in detail, with some recent experimental reports being cited:

“These two electronic-structure mechanisms will lead to the photoluminescence emission of MoS₂ from visible range to the near-infrared spectrum, including the telecommunication range at 1.55 μm [10, 12], as well as to the improved electrical property of Ln-MoS₂ [11, 15], making Ln-MoS₂ promising for optoelectronic materials and nanodevices.”